# Decreased content of ascorbic acid (vitamin C) in the brain of knockout mouse models of Na+,K+-ATPase-related neurologic disorders

**Keiko Ikeda** [1,2]*, **Adriana A. Tienda**[3], **Fiona E. Harrison**[3], **Kiyoshi Kawakami**[2]

**1** Department of Clinical Research, Group of Pathophysiology, National Hospital Organization Murayama Medical Center, Gakuen, Musashi-Murayama, Tokyo, Japan, **2** Division of Cell Biology, Center for Molecular Medicine, Jichi Medical University, Yakushiji, Shimotsuke, Tochigi, Japan, **3** Department of Medicine, Vanderbilt University Medical Center, Nashville, Tennessee, TN, United States of America

* ikeda-k@murayama-hosp.jp

**Data Availability Statement:** The data underlying this study are available on Dryad (https://doi.org/10.5061/dryad.fxpnvx0qm).

## Abstract

Na+,K+-ATPase is a crucial protein responsible for maintaining the electrochemical gradients across the cell membrane. The Na+,K+-ATPase is comprised of catalytic α, β, and γ subunits. In adult brains, the α3 subunit, encoded by *ATP1A3*, is predominantly expressed in neurons, whereas the α2 subunit, encoded by *ATP1A2*, is expressed in glial cells. In foetal brains, the α2 is expressed in neurons as well. Mutations in α subunits cause a variety of neurologic disorders. Notably, the onset of symptoms in *ATP1A2*- and *ATP1A3*-related neurologic disorders is usually triggered by physiological or psychological stressors. To gain insight into the distinct roles of the α2 and α3 subunits in the developing foetal brain, whose developmental dysfunction may be a predisposing factor of neurologic disorders, we compared the phenotypes of mouse foetuses with double homozygous knockout of *Atp1a2* and *Atp1a3* (α2α3-dKO) to those with single knockout. The brain haemorrhage phenotype of α2α3-dKO was similar to that of homozygous knockout of the gene encoding ascorbic acid (ASC or vitamin C) transporter, SVCT2. The α2α3-dKO brain showed significantly decreased level of ASC compared with the wild-type (WT) and single knockout. We found that the ASC content in the basal ganglia and cerebellum was significantly lower in the adult *Atp1a3* heterozygous knockout mouse (α3-HT) than in the WT. Interestingly, we observed a significant decrease in the ASC level in the basal ganglia and cerebellum of α3-HT in the peripartum period, during which mice are under physiological stress. These observations indicate that the α2 and α3 subunits independently contribute to the ASC level in the foetal brain and that the α3 subunit contributes to ASC transport in the adult basal ganglia and cerebellum. We propose that decreases in ASC levels may affect neural network development and are linked to the pathophysiology of *ATP1A2*- and *ATP1A3*-related neurologic disorders.

**Funding:** KI, 18K06968, Japan Society for the Promotion of Science (JSPS) KAKENHI, https://www.jsps.go.jp/english/e-grants/data/kakenhi_pamph_e.pdf, NO.

**Competing interests:** The authors have declared that no competing interests exist.

## Introduction

Na$^+$,K$^+$-ATPase (sodium-potassium adenosine triphosphatase, or Na$^+$-K$^+$ pump) is located in the membranes of neurons as well as other cells that utilize energy from hydrolysis of one molecule of ATP to move three sodium ions (Na$^+$) out of the cell in exchange for two potassium ions (K$^+$) moving inwards. The Na$^+$,K$^+$-ATPase plays an important role in maintaining (1) ionic concentrations inside and outside of the cell, (2) the osmotic balance of the cell, leading to cell volume control, and (3) cell signal transduction [1]. The ionic gradients across the cell membrane formed by the pump are used to maintain membrane potential and to generate electrical impulses in excitable cells. The gradients are also used to move neurotransmitters, glucose, amino acids, and ions across the cell membrane. The Na$^+$,K$^+$-ATPase is minimally comprised of a large multi-pass transmembrane unit (the catalytic α subunit), a heavily glycosylated β subunit, and, sometimes a regulatory γ subunit. There are four α isoforms (α1 - α4), each of which is encoded by a different gene, in humans and rodents. The human α3 subunit, encoded by *ATP1A3*, is the predominant α subunit expressed in adult neurons [2–4]. The human α2 subunit, encoded by *ATP1A2*, is expressed in astrocytes and oligodendrocytes in the adult brain [5]. Therefore, it is widely accepted that α3 is the neuronal type and α2 is the glial type. Contrary to this idea, we and others have reported that in mice, α2 is also expressed in neurons at birth [6,7].

Sporadic or inherited mutations in *ATP1A2* and *ATP1A3* have been identified and associated with various neurologic disorders. *ATP1A3*-related neurologic disorders have been shown to cause multiple distinct phenotypes: rapid-onset dystonia-parkinsonism (RDP), alternating hemiplegia of childhood (AHC), and cerebellar ataxia, areflexia, pes cavus, optic atrophy, and sensorineural hearing loss (CAPOS) syndrome [8–11]. Although most patients with *ATP1A3* mutations exhibit one of these phenotypes, some individuals show intermediate, atypical, or combined features of two or more of these major phenotypes. For example, early infantile epilepsy with encephalopathy (EIEE), recurrent encephalopathy with cerebellar ataxia (RECA), and fever-induced paroxysmal weakness and encephalopathy (FIPWE) are also caused by *ATP1A3* mutations [12–14]. Owing to this heterogeneity of symptomatic manifestations, sometimes even within the same family, *ATP1A3*-related neurologic disorders are now considered a clinical continuum and have a wide range of severity, age of onset, and progression of different signs and symptoms [15,16].

Heterozygous mutations in *ATP1A2* are found in patients with familial hemiplegic migraine type 2 (FHM2), which is an autosomal dominant form of migraine with aura [17]. There is increasing evidence that *ATP1A2* mutations are also related to epilepsy, emotional disorders [18–23], and AHC, although the number of cases of AHC caused by *ATP1A2* is small compared to that caused by *ATP1A3* [24,25]. Furthermore, heterozygous variants of *ATP1A2* have been reported in patients with early-onset stroke [26] and hypokalemic periodic paralysis [27]. The most severe clinical presentation of the *ATP1A2* variant/mutation is homozygous truncation of *ATP1A2*, which results in severe brain developmental anomalies followed by neonatal or early infantile death [28,29]. Thus, the phenotype of *ATP1A2* variants/mutations is also now considered to be a growing spectrum [30], and some of the pathophysiological outcomes of *ATP1A2* mutations seem to overlap with those of *ATP1A3* mutations.

We hypothesized that developmental dysfunction of the α2 and/or α3 subunits might be a precondition for symptomatic manifestation of these neurologic disorders. Therefore, it is important to determine whether the α2 and α3 subunits, both of which are expressed in developmental neurons, play complementary or independent roles during brain development. To address this issue, we crossed *Atp1a2/Atp1a3* double heterozygous knockout mice to generate *Atp1a2/Atp1a3* double homozygous knockout foetuses. We found that both α2 and α3

subunits contribute to the transport of ascorbic acid (ASC or vitamin C) into the foetal brain through distinct pathways. We also found that the ASC content in the basal ganglia and the cerebellum was significantly lower in adult brains of *Atp1a3* heterozygous knockout mice. Interestingly, we observed a significant decrease in the ASC level in these regions of *Atp1a3* heterozygous knockout mice in the peripartum period, during which time mice are under physiological stress. This is the first report indicating that both the α2 and α3 subunits of Na$^+$, K$^+$-ATPase play important roles in ASC uptake in the brain during development. These observations suggest that the contributions of the α2 and α3 subunits to the ASC level in the foetal brain and of the α3 subunit to the adult basal ganglia and cerebellum might be the underlying mechanisms of the pathophysiology of *ATP1A3*- and *ATP1A2*-related neurologic disorders.

## Materials and methods

### Ethics statement for the animal experiments

All experimental protocols described in the present study were carried out in a humane manner after receiving approval from and the from the Institutional Animal Experiment Committee of Jichi Medical University (No. 20–43) and in accordance with the institutional regulations of the Animal Experiments and Fundamental Guideline for Proper Conduct of Animal Experiments and Related Activities in Academic Research Institutions under the jurisdiction of the Japanese Ministry of Education, Culture, Sports, Science and Technology (MEXT), which operate in accordance with Law No. 105 of the Japanese Government for the care and use of laboratory animals. All efforts were made to minimize the number of animals used and their suffering.

### Animals

Two lines of *Atp1a2*-deficient heterozygous mice (*Atp1a2-N$^{+/-}$* and *Atp1a2-C$^{+/-}$*) and *Atp1a3*-deficient heterozygous mice (*Atp1a3$^{+/-}$*) were established as described previously [6,31,32]. *Atp1a2-N$^{+/-}$* and *Atp1a2-C$^{+/-}$* correspond to *Atp1a2$^{+/KOE2}$* and *Atp1a2$^{+/KOE21}$*, respectively, in the review by Isaksen and Lykke-Hartmann [33]. *Atp1a3$^{+/-}$* corresponds to alpha3$^{+/deltaE2-6}$ in the review by Holm and Lykke-Hartmann [34]. These mice were backcrossed to a C57BL/6J$_{Jcl}$ (CLEA Japan, Tokyo, Japan) background for 12 to 20 generations. Mice were housed under a 12-hr light/dark cycle (lights on from 7:00 AM to 7:00 PM) in a temperature-controlled room (22±2˚C). Food and drinking water (no additional ASC) were provided ad libitum. Food was a standard pelleted rodent diet (CLEA rodent diet CE-2, CLEA Japan), and the calories and nutrient components (per 100 g) were as follows: 344.2 kcal; crude protein, 25.13%; crude fat, 4.92%; Ca, 1.10 g; P, 1.09 g; Mg, 0.32 g; K, 1.01 g; Mn, 11.61 mg; Fe, 31.02 mg; Cu, 0.80 mg; Zn, 5.76 mg; Na, 0.35 mg; retinol, 1515 IU; Vitamin B$_1$, 1.9 mg; Vitamin B$_2$, 1.4 mg; Vitamin B$_6$, 1.3 mg; Vitamin B$_{12}$, 6.4 μg; Vitamin C (ASC) 25 mg; Vitamin D$_3$, 220 IU; Vitamin E, 6.2 mg; pantothenic acid, 3.6 mg; niacin, 18.1 mg; folic acid, 0.2 mg; choline, 190 mg; biotin, 44.1 μg; and inositol, 613 mg. Female and male heterozygous knockout mice were crossed, and F1 neonates were used for further analyses. Caesarean section was performed on embryonic day (E) 18.5–19.0.

### Measurement of ASC content

The mice were sacrificed by quick decapitation under deep anaesthesia using inhaled isoflurane. The whole brains or dissected regions of the brain (the cortex including the hippocampus, basal ganglia, and cerebellum) were immediately removed from foetuses/neonates or 2- to 4-month-old adult mice (female and male), respectively, quickly frozen in liquid nitrogen, and

stored at -80˚C until needed. Frozen samples were suspended in 500 µl of a solution of 0.1 M phosphate buffer (pH 2.5) plus 10 mg/L EDTA-2Na and 20 µg 3,4-dihydroxybenzylamine hydrobromide (DHBA, internal standard), and then 400 µl of methanol was added. The samples were homogenized for 1 min using an ultrasound sonicator and left on ice for 30 min. After centrifugation (20,000 x g, 15 min, 4˚C), the supernatants were diluted with 0.1 M phosphate buffer (pH 3.5) plus 1 mM EDTA-2Na. A 20 µl of filtered diluent (passed through a 0.22-µm filter) was applied to a separation column (150×3.0 mm, Eicompak SC-5ODS). High-performance liquid chromatography (HPLC) with electrochemical (EC) detection was performed at +0.8 V vs. Ag/AgCl with a flow rate of 450 µl/min (HITEC-500, EICOM, Kyoto, Japan). Quantitation of ASC content in each sample was achieved by comparing peak heights with those obtained after injections of known quantities of ASC and DHBA (standard, in 20 µl, 40 pg and 200 pg, respectively). The data were analysed using the EPC-300 system at a 10/sec sample speed (EICOM). The values were normalized to the weights of the specimen (whole brain or parts of the brain).

## RNA extraction, reverse transcription, and quantitative PCR

Frozen whole brains were homogenized in 1 ml of ISOGEN reagent (Nippon Gene, Tokyo, Japan). Total RNA was extracted from the tissue according to the manufacturer's suggested protocol. The RNA was resuspended in water, and the concentration was determined by $A_{260}$ measurement using a DU spectrophotometer (Beckman Coulter, CA, USA). First strand cDNA was synthesised from 1 µg of each total RNA in 20 µl of reaction buffer using the Prime-Script II 1st Strand cDNA Synthesis Kit (Takara Bio, Shiga, Japan). The resulting cDNA products obtained from 1 µl of reaction buffer were used for quantitative PCR on a StepOne Real-time PCR system using TaqMan assays (Applied Biosystems, Massachusetts, USA) to measure the expression of mouse *Slc23a2*, which encodes sodium-dependent vitamin C transporter 2 (SVCT2) (Mm00497751_m1) and glyceraldehyde-3-phosphate dehydrogenase (Gapdh, Mm99999915_g1) as an internal control. Three independent experiments were performed, and in each experiment, the value of one of the wild-type (WT) samples was set as 1.0. The average relative expression values were as follows: WT, 0.979±0.019 (n = 6); α2N/α3-dKO, 1.061±0.101 (n = 4); α2C/α3-dKO, 0.828±0.082 (n = 3); α2N-KO, 0.892±0.060 (n = 4); α2C-KO, 0.922±0.101 (n = 5); and α3-KO 0.892±0.104 (n = 5).

## Measurement of malondialdehyde (MDA) and protein sulfhydryl (SH) levels

Frozen brain regions (the cortex including hippocampus, basal ganglia, and cerebellum) prepared as described above were used. Analyses of MDA and SH levels were performed as previously described [35–38] with some modifications. Briefly, each sample was homogenized in a small volume of 1 ml of 5% trichloroacetic acid. The volume was made up to 1 ml, and the samples were centrifuged at 13,000 rpm at room temperature for 5 min. Then, 250 µl of each sample was mixed with the same volume of 0.02 M thiobarbituric acid for 35 min at 95˚C, and the reaction was terminated by incubation at 4˚C for 10 min. MDA levels in 200 µl samples of the final product were quantified in duplicate by measuring fluorescence in a 96-well plate (Synergy H4 plate reader; excitation at 515, emission at 553). The final concentration was calculated relative to the wet weight with reference to a standard curve generated from a stock solution of 50 pmol/µl MDA that was treated in an identical manner to the samples. For SH level determination, the samples were homogenized in 10 volumes of sodium phosphate buffer (pH 7.4) with 140 mM KCl and a protease inhibitor pellet. Each sample was then centrifuged at 750 x g at 4˚C for 10 min. The supernatant was diluted 1:1 in homogenization buffer.

Protein content was calculated using a standard BCA assay, and 10 μl of sample was reacted with 6 μl of 10 mM DTNB in 0.2 M potassium phosphate solution (pH 8) with an additional volume of 200 μl of 1x PBS (pH 7.4) containing 1 mM EDTA. The samples were incubated at room temperature in the dark for 30 min, and the absorption was then measured at 412 nm.

## Anatomical and histological examination

Histochemistry was performed on 30-μm cryosections as described previously [6], followed by haematoxylin-eosin staining. Images of sections were obtained at 4×, 10×, or 20× magnification with a conventional microscope (BX51, Olympus, Tokyo, Japan). Foetuses and their brains were imaged using stereomicroscopy (SZX7, Olympus).

## Statistical analyses

The data were analysed using GraphPad Prism version 6. MDA and SH data were analysed by unpaired t-test. All other data were evaluated by either univariate ANOVA, or two-way ANOVA as appropriate, followed by Tukey's or Sidak's multiple comparison test, respectively. Significance was set at $p < 0.05$. The P or F without a subscript "s" refers a single ANOVA response and the P or F with a subscript "s" refers to two post hoc comparisons.

## Generation of *Slc23a2*-KO mice

*Slc23a2*-KO mice were generated by a modified CRISPR/Cas9 gene editing system [39]. This system enables homologous recombination-mediated targeted loxP sequence integrations by using left and right single-guide RNA/Cas9 expression vectors and another vector with long double-stranded donor DNA spanning both the left and right CRISPR target sequences. The vectors designated pX330-Slc23a2-left and pX330-Slc23a2-right, which express the single-guide RNA/Cas9, were made by Seiya Mizuno. Briefly, the left guide RNA sequence (5′-GGGCTAGCTCCATCGGACAT-3′) and the right guide RNA sequence (5′-TCCATAGCCCCGT AACAGTG-3′) were inserted into the entry site of the pX330 vector as previously reported [40]. The donor vector, named pflox-Slc23a2-dsDonor, which was based on the pflox vector backbone, was also generated by Seiya Mizuno and was designed to flank the exon containing the entire open reading frame of *Slc23a2* with loxP sites to allow for conditional removal by Cre recombinase. pflox-Slc23a2-dsDonor contained a left homology arm (1,290 bp) and a right homology arm (1,437 bp) that flanked the left and right CRISPR target sequences, respectively. For microinjection, these 3 vectors were isolated with a FastGene Gel/PCR Extraction Kit (Nippon Genetics, Tokyo, Japan) and diluted to 5 ng/μl (pX330- Slc23a2-left/right vectors) or 10 ng/μl (pflox-Slc23a2-dsDonor) with deionized distilled water. After filtration with a 0.22-μm filter (Millex-GV; Merck Millipore, Billerica, Massachusetts, USA), these vectors were microinjected into the male pronuclei of fertilized oocytes according to standard protocols [40]. Surviving 1-cell embryos were transferred into the oviducts of pseudopregnant ICR females. Slc23a2-KO mice (strain deposited in the RIKEN BioResource Research Center) and Slc23a2-floxed mice were generated as previously described [40,41]. Because all the Slc23a2-floxed mice were found to be random integrations in non-homologous chromosomes, we used only Slc23a2-KO mice for the subsequent analyses. All mice were on a C57BL/6 genetic background. Genotyping was performed using the following primer sets: Set 1, 5'-GGTATTT GTAAAAGCCTTGGCCAGAATG-3' and 5'-ATGGTCCACACCACGATCTAGCAGTTAC-3' to produce a 7,440-bp fragment for the WT allele and a 3,616-bp fragment for the deletion allele; Set 2, 5'-GGTATTTGTAAAAGCCTTGGCCAGAATG-3' and 5'-TACTCAGTGTCCCTCTAGCTGC CTCATC-3' to produce a 5,831-bp fragment for the WT allele and a 2,007-bp fragment for the deletion allele); and Set 3, 5'-AGCTCTCTTGCAGAGACCCTGGACAG-3' and 5'-CTGCATCCAA

ATGTTGTCTGCAGCAATG-3' to produce a 2,234-bp fragment for the WT allele and no DNA bands for the deletion allele). Female and male heterozygous knockout mice were crossed, and F1 neonates were used for further analyses.

## Results

### Extensive brain haemorrhage in *Atp1a2* and *Atp1a3* double homozygous knockout foetuses

We previously reported two *Atp1a2* heterozygous knockout mouse lines: *Atp1a2-N*[+/-] and *Atp1a2-C*[+/-] (Table 1) [6,31]. We also reported on the *Atp1a3* heterozygous knockout mouse line (*Atp1a3*[+/-]) [32]. Although the primary causes of the human *ATP1A2*- and *ATP1A3*-related neurologic disorders described above are missense mutations, these heterozygous knockout (deletion) mice have been shown to exhibit the phenotypes of FHM2 (*Atp1a2*[+/-]) [42] and RDP but not AHC (*Atp1a3*[+/-]) [43,44]. Brains isolated from homozygous knockout neonates of either line (*Atp1a2-N*[-/-] and *Atp1a2-C*[-/-]) showed the complete absence of the $Na^+$, $K^+$-ATPase α2 subunit protein [6,31], and the α3 subunit protein was completely absent in the brains of homozygous knockout (*Atp1a3*[-/-]) neonates [32,45]. *Atp1a2-N*[-/-], *Atp1a2-C*[-/-], and *Atp1a3*[-/-] foetuses died immediately after birth due to a lack of spontaneous body movements and functional defects in the respiratory neural network that resides in the medulla oblongata [6,31,45–47]. However, no gross morphological anomalies were observed in the brains or spinal cords of *Atp1a2-N*[-/-], *Atp1a2-C*[-/-], and *Atp1a3*[-/-]. At present, there have been no reports of patients with double mutations in *ATP1A2* and *ATP1A3*. Nevertheless, to determine whether

**Table 1.** *Atp1a2* (ATPase, Na+/K+ transporting, alpha 2 polypeptide) and *Atp1a3* (ATPase, Na+/K+ transporting, alpha 3 polypeptide) knockout mice used in this paper and their phenotypes.

| Mouse name in this paper | Genetic alteratsultion | Nomenclature symbol in MGI (Mouse Genome Informatics) MGI URL | MGI ID | Name in [33,34] | Major observations | Reference(s) |
|---|---|---|---|---|---|---|
| *Atp1a2-N*[+/-] or α2N-HT | Deletion targeting exon 2 in one allele of *Atp1a2* | Atp1a2[tm2Kwk]/Atp1a2[+] (heterozygous knockout) http://www.informatics.jax.org/allele/MGI:3522421 | MGI: 3522421 | α2[+/KOE2] | Enhanced CSD induction. | [31,42] |
| *Atp1a2-N*[-/-] or α2N-KO | Deletion targeting exon 2 in both alleles of *Atp1a2* | Atp1a2[tm2Kwk]/Atp1a2[tm2Kwk] (homozygous knockout) http://www.informatics.jax.org/allele/MGI:3522421 | MGI: 3522421 | α2[KOE2/KOE2] | Complete absence of α2 protein in the brain. High $[Cl^-]_i$ in neurons due to functional uncoupling with KCC2. Respiratory neural network dysfunction. Different phenotypes depending on the mode of delivery. | [31,46,47,49] |
| *Atp1a2-C*[+/-] or α2C-HT | Deletion targeting exon 21 in one allele of *Atp1a2* | Atp1a2[tm1Kwk]/Atp1a2[+] (heterozygous knockout) http://www.informatics.jax.org/allele/MGI:2664907 | MGI: 2664907 | α2[+/KOE21] | Enhanced fear and anxiety behaviours. Enhanced c-Fos expression in the amygdala. Obesity due to hyperphagia. Enhanced CSD induction. | [6,42,50] |
| *Atp1a2-C*[-/-] or α2C-KO | Deletion targeting exon 21 in both alleles of *Atp1a2* | Atp1a2[tm1Kwk]/Atp1a2[tm1Kwk] (homozygous knockout) http://www.informatics.jax.org/allele/MGI:2664907 | MGI: 2664907 | α2[KOE21/KOE21] | Complete absence of α2 protein in the brain. Respiratory neural network dysfunction. Degeneration of the amygdala and piriform cortex. | [6] |
| *Atp1a3*[+/-] or α3-HT | Deletion targeting exons 2–6 in one allele of *Atp1a3* | Atp1a3[tm1.1Kwk]/Atp1a3[+] (homozygous knockout) http://www.informatics.jax.org/allele/MGI:5572809 | MGI: 5572809 | α3[+/E2−6] | Increased locomotor activity. Increased dystonic response to intracerebellar kainate injections. Enhanced inhibitory neurotransmission. Stress-induced motor deficits. Lower rank in hierarchy and altered social behaviour. | [32,43,44] |
| *Atp1a3*[-/-] or α3-KO | Deletion targeting exons 2–6 in both alleles of *Atp1a3* | Atp1a3[tm1.1Kwk]/Atp1a3[tm1.1Kwk] (homozygous knockout) http://www.informatics.jax.org/allele/MGI:5572809 | MGI: 5572809 | α3[E2−6/E2−6] | Complete absence of α3 protein in the brain. Perinatal seizures. Increased monoamine content in the brain. Defect in respiratory rhythm generation. | [32,45] |

the α2 and α3 subunits play complementary or independent roles during brain development, we decided to examine double homozygous knockout mice. We generated *Atp1a2* and *Atp1a3* double heterozygous knockout mice ($Atp1a2$-$N^{+/-}Atp1a3^{+/-}$) by crossing $Atp1a2$-$N^{+/-}$ and $Atp1a3^{+/-}$. Then, male and female $Atp1a2$-$N^{+/-}Atp1a3^{+/-}$ mice were crossed to obtain double homozygous knockout mice ($Atp1a2$-$N^{-/-}Atp1a3^{-/-}$) (hereafter termed α2N/α3-dKO). The α2N/α3-dKO neonates showed slow sinus rhythm and died within one hour of birth due to a lack of spontaneous body movements, including respiratory activity. Surprisingly, all α2N/α3-dKO neonates showed extensive haemorrhage over the convex surface of the brain (Fig 1A–1C) that was sometimes

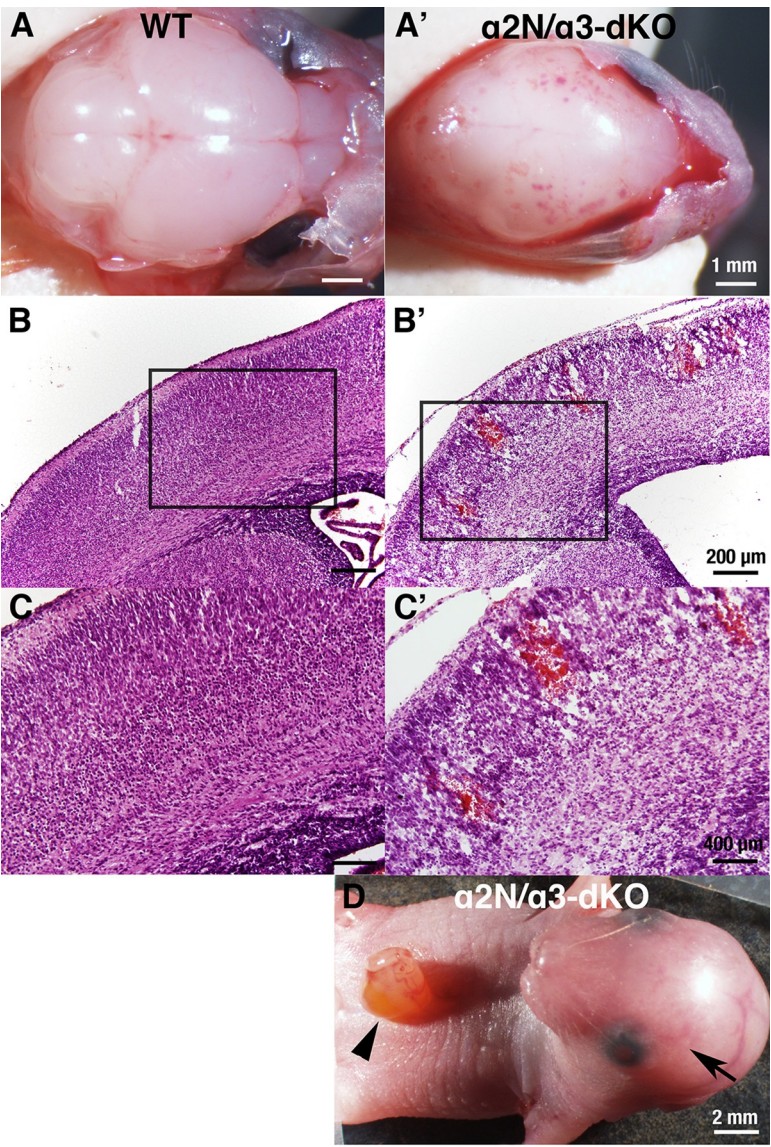

**Fig 1. Phenotype of wild-type (WT) and *Atp1a2-N*/*Atp1a3* double homozygous knockout (α2N/α3-dKO) mice. A, A'.** Gross examination of the surface of the brains of WT and α2N/α3-dKO newborn mice after removal of the cranial bones. The α2N/α3-dKO showed petechiae and ecchymoses over cerebral convexity. Scale bars, 1 mm. **B, B'.** Microscopic examination of the cortex. Intraparenchymal haemorrhage in the cortex of α2N/α3-dKO are shown. Scale bars, 200 μm. **C, C'.** Magnification of the boxed regions in B and B', respectively. Scale bars, 400 μm. **D.** Abdominal wall rupture observed in α2N/α3-dKO mice (arrowhead). Brain haemorrhage was observed through the skin and cranial bones (arrow). Scale bars, 2 mm.

visible through the skin overlying the cranium (Fig 1D, arrow). In addition, most of the α2N/α3-dKO neonates showed abdominal wall rupture (Fig 1D, arrowhead). The brain haemorrhages varied from dotted petechiae to large ecchymoses. Microscopic examination revealed widespread intraparenchymal haemorrhage in the α2N/α3-dKO mice (Fig 1B and 1C). The bleeding sites corresponded to the penetrating arteries, which are the most fragile blood vessels in the human premature foetal brain [48]. To confirm that the haemorrhage phenotype was due to combined deficiency of *Atp1a2* and *Atp1a3* genes, we generated *Atp1a2-C<sup>-/-</sup>Atp1a3<sup>-/-</sup>* mice (hereafter referred to as α2C/α3-dKO) by crossing *Atp1a2-C<sup>+/-</sup>* and *Atp1a3<sup>+/-</sup>* mice. We also observed brain haemorrhage in α2C/α3-dKO neonates, as in α2N/α3-dKO neonates (https://doi.org/10.5061/dryad.fxonvx0qm). These results indicated that combined deficiency of *Atp1a2* and *Atp1a3* caused brain haemorrhage and that the haemorrhage phenotype was independent of the strategy used for *Atp1a2* gene disruption.

## Significant decrease in ASC content in the brains of *Atp1a2/Atp1a3* double homozygous knockout mice

We noticed that the extensive haemorrhage in the brains of α2N/α3-dKO and α2C/α3-dKO foetuses closely resembled the phenotype of newborns with homozygous knockout of the ASC transporter gene (solute carrier family 23 member 2 (*Slc23a2*); formerly designated *Slc23a1*) [37,51]. *Slc23a2* is highly expressed in the brain and adrenal gland [52], where it allows ASC to accumulate against a gradient at higher concentrations than in all other organs. It has been reported that foetuses of *Slc23a2<sup>-/-</sup>* (hereafter termed SVCT2-KO) die within a few minutes of birth, exhibiting severe brain haemorrhage and a complete absence of ASC in the brain [37,51]. To address whether the haemorrhage phenotype of α2N/α3-dKO and α2C/α3-dKO mice was correlated with a lower content of ASC, we measured the ASC content in the whole brains of E18.5–19 foetuses of α2N/α3-dKO, α2C/α3-dKO, and WT littermates. ASC content in the brain was significantly decreased in all knockout foetuses examined compared with WT ($F_{5, 19}$ = 63.51, P<0.001, Fig 2A). The ASC content in the brain was decreased by approximately 50% in α2N/α3-dKO and α2C/α3-dKO compared to WT (Ps<0.001) but did not differ between the α2N/α3-dKO and α2C/α3-dKO mice (P = 0.22). The ASC content was also decreased by 26%-30% in the single homozygous knockout *Atp1a2-N<sup>-/-</sup>* (α2N-KO), *Atp1a2-C<sup>-/-</sup>* (α2C-KO), and *Atp1a3<sup>-/-</sup>* (α3-KO) compared to the WT (Ps<0.01) but did not differ among the different single homozygous knockout mice (Ps>0.96). In general, the double knockout mice exhibited a greater decrease in ASC content than the single knockout animals (Ps<0.02), although this was not significant for α2N/α3-dKO vs. α2C-KO (P = 0.085). These results indicate that *Atp1a2* and *Atp1a3* are critically and independently involved in the transport of ASC into the foetal brain.

SVCT2, which is encoded by *Slc23a2*, uses the Na<sup>+</sup> gradient across the cell membrane generated by Na<sup>+</sup>,K<sup>+</sup>-ATPase. Therefore, we decided to investigate the interaction between the *Slc23a2* and *Atp1a3* genes. First, we examined the expression level of *Slc23a2* in the foetal brain. We found that there was no significant difference in the mRNA expression of *Slc23a2* among genotypes (Fig 2B). The results indicate that decreased ASC contents in the double homozygous knockout (dKOs) and single homozygous knockout animals (KOs) were caused by functional disturbance of SVCT2, not by the decreased *Slc23a2* expression.

## Generation of *Slc23a2* knockout mice and confirmation of decreased ASC content in the brain

Next, we generated a *Slc23a2* knockout mouse line by using CRISPR-Cas9 (Fig 3A). As previously reported [37,51], SVCT2-KO (homozygous knockout foetuses) showed extensive brain

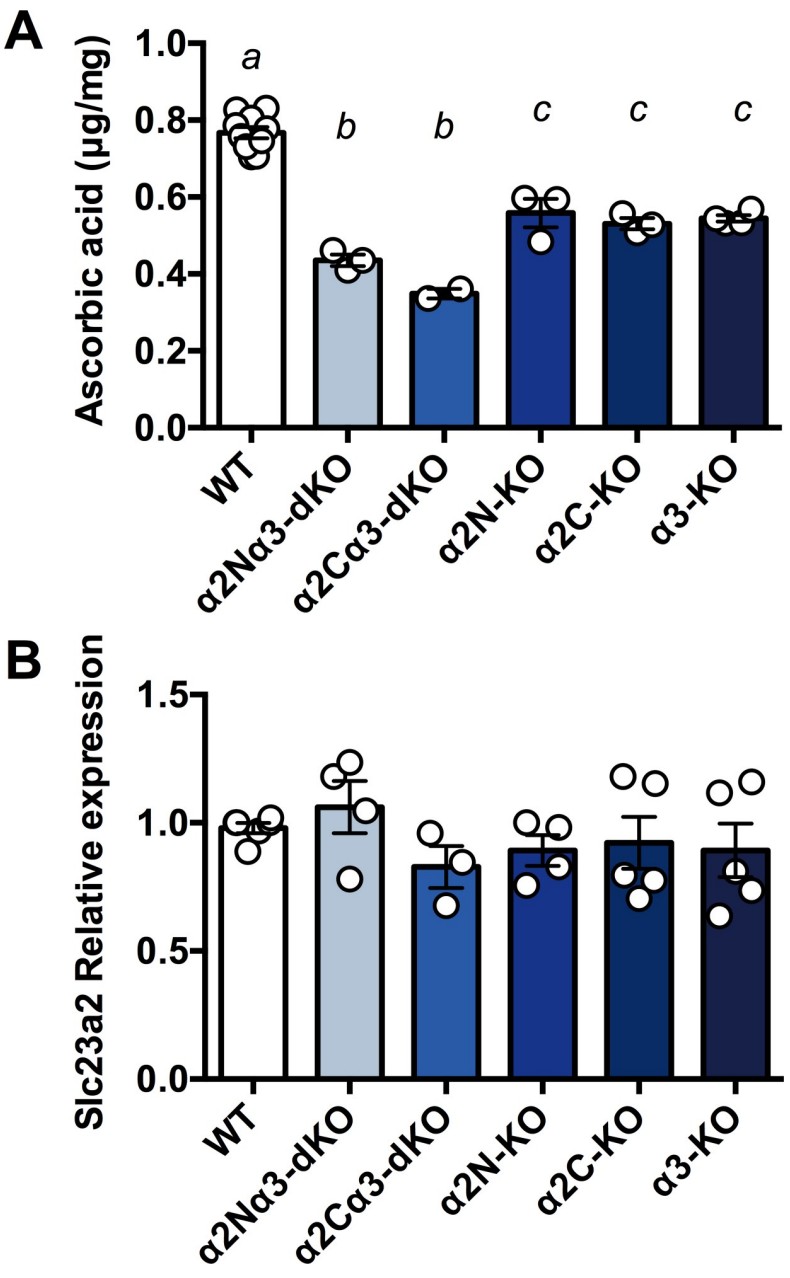

**Fig 2. Ascorbic acid (ASC) content and *Slc23a2* expression in the whole foetal brain. A.** The ASC content in the whole brains isolated from foetuses was measured using HPLC with chemical detection. α2N/α3-dKO, a double homozygous knockout of *Atp1a2N/Atp1a3*; α2C/α3-dKO, a double homozygous knockout of *Atp1a2C/Atp1a3*; α2N-KO, a homozygous knockout of *Atp1a2-N*; α2C-KO, a homozygous knockout of *Atp1a2-C*; α3-KO, a homozygous knockout of *Atp1a3*. Values for WT from separate litters were combined for analyses. Data were analysed using univariate ANOVA with Tukey post hoc follow-up analyses, corrected for multiple comparisons. Bars marked with separate letters are significantly different from each other (P<0.05), except in one case, i.e., α2Nα3-dKO vs. α2C-KO where the trend was not significant (P = 0.085). a vs. b & a vs. c P<0.001, b vs. c P<0.05. Data are presented as the mean ± SE. **B.** Relative mRNA expression of *Slc23a2* in whole brains isolated from foetuses. The expression level in one of WT brains was set as 1.0 in three independent experiments. No significant differences were observed among genotypes.

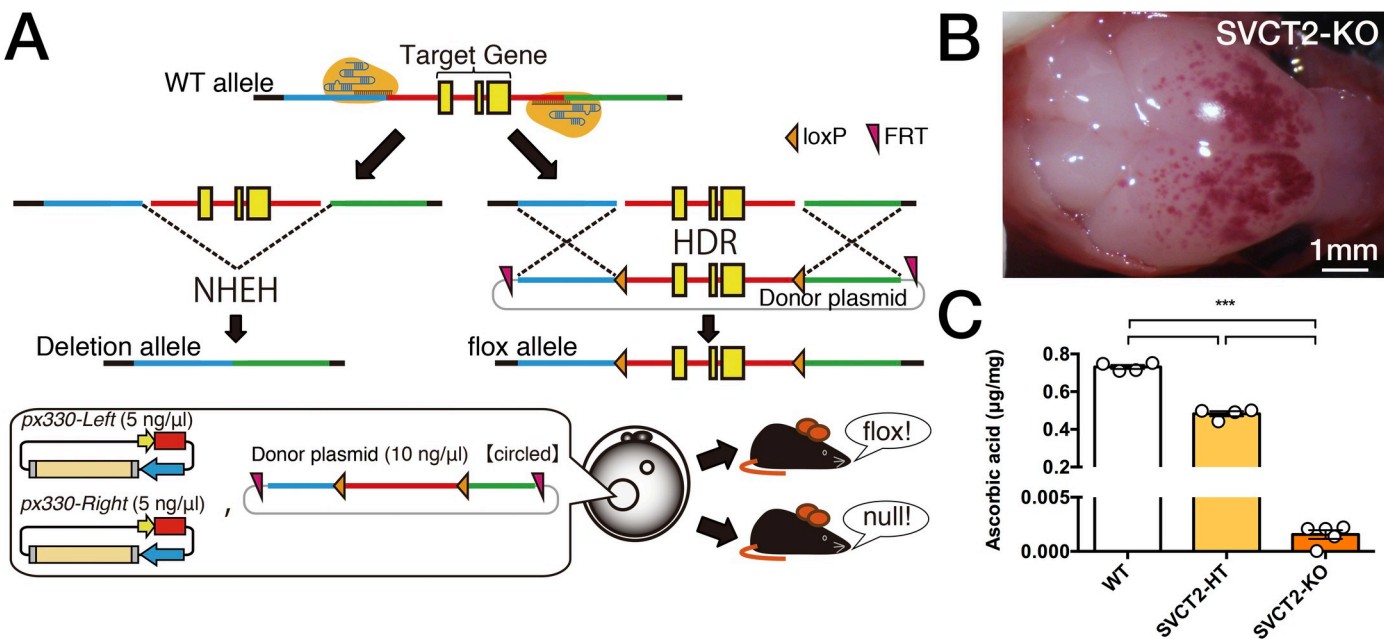

**Fig 3. Strategy used for generation of *Slc23a2*-KO mice by a modified CRISPR/Cas9 gene editing system. A.** Method used to generate *Slc23a2*-floxed and *Slc23a2*-null mice using the CRISPR/Cas9 system. **B.** Gross examination of the surface of the brain after removal of the cranial bones. The *Slc23a2* homozygous null (SVCT2-KO) showed petechiae and ecchymoses over cerebral convexity. Scale bars, 1 mm. **C.** Whole brain ASC levels measured by HPLC with chemical detection. For comparison, the ASC content in *Slc23a2* heterozygous null (knockout) (SVCT2-HT) brains is shown. Data are presented as the mean ± SE, n = 4–5 foetuses per group. The groups were all different at P<0.001 as indicated (***).

haemorrhage and died just after birth (Fig 3B). We confirmed that ASC was dramatically depleted and was almost undetectable in the whole brains of homozygous knockout foetuses ($F_{2, 10}$ = 1641, P<0.001; Fig 3C). This decrease in ASC levels in SVCT2-KO was more severe than that in α2N/α3-dKO and α2C/α3-dKO (Fig 2). The average raw values of whole brain weight, which was not significantly different among the genotypes, were as follows: WT, 87.1 ± 4.19 mg (n = 4); SVCT2-HT, 86.5 ± 0.90 mg (n = 4); and SVCT2-KO, 82.3 ± 3.62 mg (n = 5). We also observed that the death of SVCT2-KO mice occurred immediately after birth, similar to previous reports [37,51]. Next, we crossed *Atp1a3*[+/-] mice (hereafter termed α3-HT) and *Slc23a2* heterozygous mice (hereafter termed SVCT2-HT) to obtain double heterozygous mice (α3-HT/SVCT2-HT).

## Significant decrease in ASC content in the basal ganglia and cerebellum of adult α3-HT

We isolated the cortex, basal ganglia, and cerebellum from adult mouse brains (2–4 months old), measured the ASC content in α3-HT and their WT littermates, and compared the magnitude of ASC depletion to SVCT2-HT and α3-HT/SVCT2-HT. SVCT2-HT led to the expected ~35% decrease in brain ASC levels in all areas tested (main effect of SVCT2 genotype $Fs_{1, 66}$ = 283.0, P<0.001). In the cortex, there was no additional impact of *Atp1a3* genotype (Fs>2.064, Ps>0.15 for main effect of *Atp1a3* genotype and interaction, Fig 4A–4C). Interestingly, Tukey's post hoc analyses confirmed a small effect of *Atp1a3* knockout alone on ASC levels in the basal ganglia and even more so in the cerebellum, leading to an approximately 10% decrease in ASC content compared to that in the same regions in WT (basal ganglia: main effect of *Atp1a3* $F_{1, 66}$ = 3.773, P = 0.056; interaction $F_{1, 66}$ = 3.305, P = 0.073; cerebellum: main effect of *Atp1a3* $F_{1, 66}$ = 22.88, P<0.001, Fig 4B and 4C). There was no further decrease in ASC

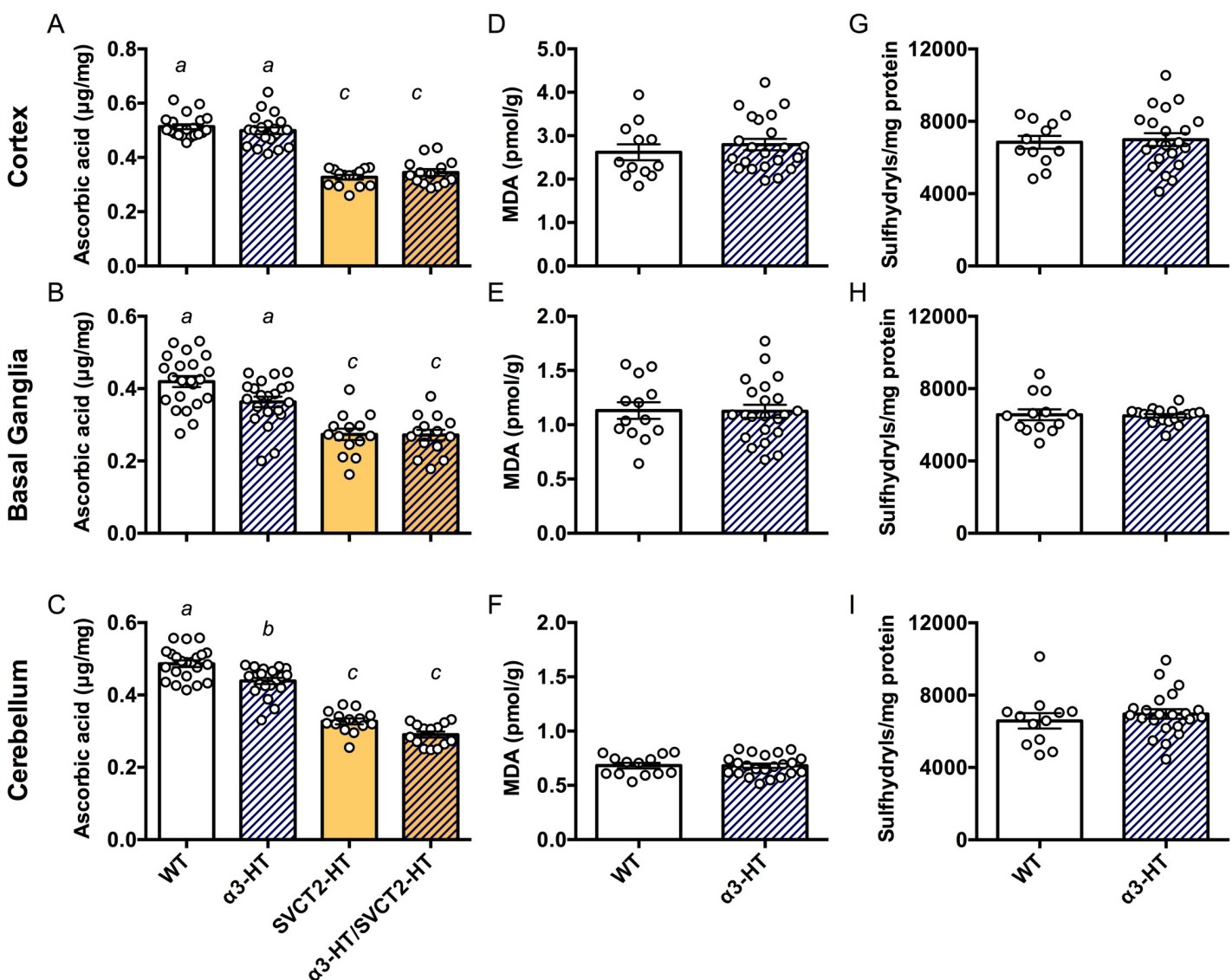

**Fig 4. Contents of ASC and oxidative stress marker molecules in parts of adult mouse brains.** The ASC content in different regions (the cortex, basal ganglia, and cerebellum) in brains isolated from 2- to 4-month-old adult mice of WT, α3-HT (*Atp1a3* heterozygous knockout), SVCT2-HT, and α3-HT/SVCT2-HT, measured by HPLC with chemical detection. **A.** Cortex. **B.** Basal ganglia. **C.** Cerebellum. Bars with different letters are significantly different, as determined by Tukey's post hoc test. a vs. b, P<0.05; a vs. c and b vs. c, P<0.001. Oxidative stress was estimated by measuring malondialdehyde (MDA) and sulfhydryl levels, in the cortex (**D** and **G**), basal ganglia (**E** and **H**), and cerebellum (**F** and **I**). Statistical analysis by Student's t-test revealed no significant difference in MDA and sulfhydryl levels between WT and α3-HT mice. N = 14–22 mice per group.

content in the double heterozygous knockout mice, suggesting that where the deletion mutation in *Slc23a2* (SVCT2) impairs the function of a whole transport pathway, the consequence of additional deletion mutations of *Atp1a3* are likely masked.

Next, we measured the levels of markers of oxidative stress in the brain, including malondialdehyde (MDA), a product of lipoperoxidation, and sulfhydryl (SH) groups (thiols), which are both sensitive markers of an altered oxidation state due to ASC deficiency [35,53]. The modest decrease in ASC content in the adult α3-HT brain (basal ganglia and cerebellum) was not associated with increases in MDA or SH groups in the cortex, basal ganglia, or cerebellum compared to the WT (Fig 4D–4I). The results highlight the extent of ASC decrease in α3-HT

mice did not drive increased oxidative stress and further suggest that the ASC decrease was not generated by an excess of reactive oxygen species (ROS). Instead, we propose that the change was due to some other facet of ASC transporter function coupled with the α3 subunit of $Na^+,K^+$-ATPase.

### Significant decrease in ASC content in the cerebellum of α3-HT in the peripartum period

*ATP1A3*-related neurologic disorders often manifest due to physical and/or psychological triggers, including childbirth [15]. The α3-HT mice used here are usually asymptomatic and only manifest modest neurologic defects upon physiological stress [43,44]. Other knock-in mice that harbour mutations observed in AHC patients or single point mutations induced by mutagenesis screening exhibit spontaneous neurologic defects such as seizures and hemiplegia and frequently show failure to thrive [54–57]. Therefore, the α3-HT line is a suitable model for investigating trigger-induced neurologic defects. Actually, we observed transient dystonic spells in female α3-HT mice after delivery (n = 2, https://doi.org/10.5061/dryad.fxonvx0qm). To examine whether physiological stress affects ASC levels in the brain, we measured ASC content in brain regions (the cortex, basal ganglia, and cerebellum) of peripartum female mice that were either at full-term pregnancy or lactating within 12 hours of delivering a litter. Neither genotype nor pregnancy impacted ASC content in the cortex (Fs<0.64, Ps>0.43) (Fig 5). As previously observed in adult α3-HT, ASC levels were modestly decreased in the basal ganglia (main effect of α3-HT genotype, $F_{1,\,22} = 22.4$, P<0.001), but this effect was increased by pregnancy (genotype x pregnancy interaction, $F_{1,\,22} = 4.33$, P<0.05). Similarly, α3-HT exhibited overall lower ASC levels in the cerebellum ($F_{1,\,22} = 17.81$, P = 0.0001), which were also further decreased by pregnancy (genotype x pregnancy interaction, $F_{1,\,22} = 4.39$, P<0.05).

## Discussion

### $Na^+,K^+$-ATPase α subunits play a critical role in transporting ASC into the brain

The $Na^+$ and $K^+$ concentration gradient across the cell membrane formed by $Na^+,K^+$-ATPase is crucial for physiological processes in neurons and astrocytes to maintain the resting membrane potential and to generate action potential. In addition, the gradient provides the driving

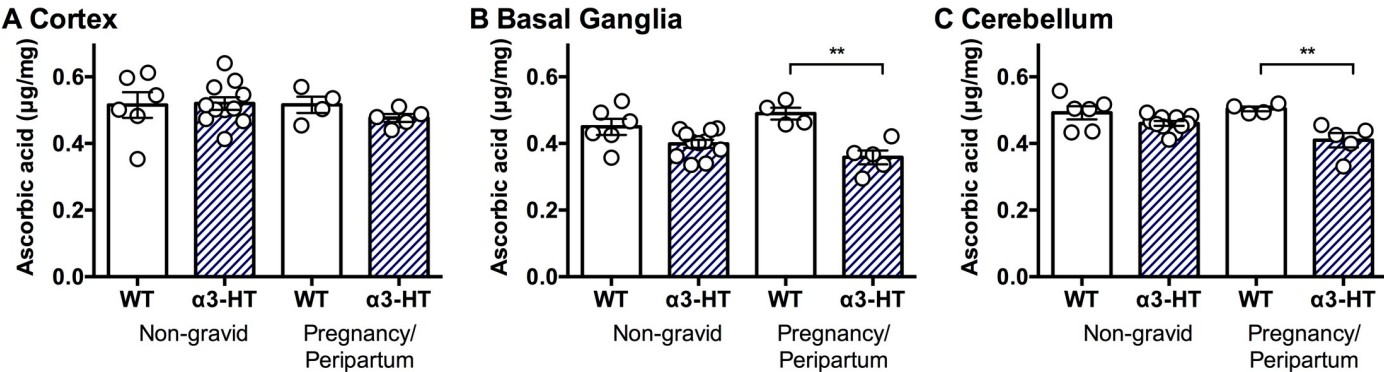

**Fig 5. ASC Content in the brain regions isolated from non-gravid or peripartum WT and α3-HT mice.** The ASC content in the different regions (the cortex, basal ganglia, and cerebellum) of brains isolated from non-gravid (control) and pregnancy/peripartum (under physiological stress) mice was measured by HPLC with chemical detection. **A.** Cortex. **B.** Basal ganglia. **C.** Cerebellum. Data from 4 to 11 mice per group were analysed by 2-way ANOVA with genotype and pregnancy as independent variables followed by Sidak's multiple comparisons test. Differences between selected pairs are marked with ** (P<0.01). The data are presented as the mean ± SE.

force for the function of several secondary active transporters that transport neurotransmitters, calcium ions, chloride ions, and nutrients (glucose and amino acids), which are all essential for maintaining adequate neurotransmission [31,58]. Here, we showed that the $Na^+$ gradient formed by $Na^+,K^+$-ATPase is essential for transporting ASC into the brain during foetal development by examining double homozygous knockout foetuses of the α2 and α3 subunits of $Na^+,K^+$-ATPase gene (α2N/α3-dKO and α2C/α3-dKO). The brain haemorrhage phenotype and decrease in ASC content in the brains of these dKOs strongly resembled those in mice with homozygous knockout of *Slc23a2* (SVCT2-KO or SVCT2$^{-/-}$) [37,51]. The post-birth death of SVCT2-KO mice was previously presumed to be caused by respiratory failure [51]. We also observed sinus rhythm but a lack of spontaneous respiration in our SVCT2-KO, similar to homozygous $Na^+,K^+$-ATPase α2 and α3 knockout mice (α2N-KO and α3-KO). Using brainstem-spinal cord preparations, in which most of the respiratory generator neural network is preserved, we used electrophysiological approaches to assess functional defects in synaptic transmission in the respiratory rhythm generator neural complexes on the ventral surface of the medulla oblongata in α2N-KO and α3-KO foetuses [6,31,45–47]. Further analyses of the respiratory neural networks of SVCT2-KO mice using brainstem-spinal cord preparations are ongoing in an attempt to uncover the role of ASC and SVCT2 in functional respiratory neural network formation during development and the cause of death.

## Insights into the decrease of ASC in the brain and *ATP1A2*- and *ATP1A3*-related neurologic disorders

ASC is a vital antioxidant molecule in the brain (reviewed in [52]). Intracellular ASC helps maintain the integrity and function of several processes in the brain, such as neuronal differentiation/maturation, myelin formation, synthesis of neurotransmitters (catecholamines), modulation of dopaminergic and glutamatergic neurotransmission, and antioxidant protection [52,59,60]. Allosteric modulation of GABA receptors by ASC has also been reported [61]. To meet the demand of the brain, ASC concentrations in human cerebrospinal fluid (CSF) are higher than those in plasma, and neurons and glia participate in active transport systems that achieve intracellular ASC levels in the millimolar range [52,62]. The steep ASC concentration gradient between the blood and neurons is formed first via the transport of ASC into the CSF through ependymal cells (which express the $Na^+,K^+$-ATPase α2 subunit) [63] and second via transport of ASC into neurons (which express the $Na^+,K^+$-ATPase α3 subunit). Our results showing a further decrease in ASC content in α2N/α3-dKO or α2C/α3-dKO compared to single knockout mice (α2N-KO, α2C-KO, or α3-KO) (Fig 2A) strongly support the existence of these two distinct pathways in the foetal mouse brain. Unlike in adult α3-HT, we did not observe a decrease in ASC content in brain regions isolated from adult α2-HT (*Atp1a2N*$^{+/-}$) compared to their WT littermates (https://doi.org/10.5061/dryad.fxpnvx0qm). We hypothesize that in the adult brain, the α2 subunit colocalizes with SVCT2 in more restricted cells, such as ependymal cells [63], which compose a relatively small region of the adult whole brain. Alternatively, the function of the α2 subunit may be compensated by the ubiquitously expressed α1 subunit during brain development [64,65].

Because the highest concentrations of ASC in the body are found in the brain and neuroendocrine tissues, as mentioned above, it is not surprising that ASC deficiency in the brain is related to various neurological diseases in humans [66] and in model animals. The following evidence provides examples of this phenomenon in rodents: depletion of ASC in guinea pig results in brain anomalies [67]; guinea pigs with combined deficiency of ASC and vitamin E show paralysis and neuronal death [68]; gulo$^{-/-}$ mice, which are unable to synthesize ASC, show sensorimotor deficits [69]; gulo$^{-/-}$ mice with severe ASC deficient intake exhibit motor

deficits [70]; gulo[-/-] mice exhibit exaggerated hyperactivity in response to a dopaminergic agonist [71]; gulo[-/-]SVCT2[+/-] mice with vitamin C and E deficiency show motor defects [72]; SVCT2[+/-]/APP/PSEN1 mice with low ASC levels exhibit seizures and cognitive impairment [73–75]; and AKR1A[-/-] mice with ASC insufficiency show impaired spatial memory formation [76].

Seizures are commonly observed in *ATP1A2*- and *ATP1A3*-related neurological disorders [15,77], and in a mouse model of Alzheimer's diseases, seizures are exaggerated by low ASC levels [74,75]. The excitatory neurotransmitter glutamate causes depolarization of mitochondria and increases ROS production, while ASC inhibits ROS and limits excitotoxic cell death [78], which may make neurons particularly sensitive to ASC depletion. We reported neuronal hyperexcitation, which resulted in robust apoptosis (excitotoxicity), in the amygdala/piriform cortex of *Atp1a2* homozygous knockout mice (α2C-KO) [6] and neuronal hyperexcitation in the piriform cortex, basal ganglia, and cerebellum of *Atp1a3* homozygous knockout mice with epileptic seizures (α3-KO) [32]. Enhanced fear and anxiety behaviours and increased excitation of amygdala neurons were observed in α2C-HT [6]. A close relationship between glutamate signalling, ROS, and *ATP1A2/ATPA3*-related neurological disorders has also been suggested [79]. Therefore, it is likely that decreased levels of ASC may contribute to the symptomatic manifestation of *ATP1A2/ATP1A3*-related neurologic disorders.

## A decrease in ASC content under stress may be related to the symptoms of *ATP1A3*-related neurologic disorders

Very recently, it was found that patients with AHC frequently show congenital and subsequent cerebellar hypoplasia and atrophy [80–82] and that FHM2 patients show biochemical changes in the cerebellum, as measured by proton-nuclear magnetic resonance spectroscopy [83]. Together with our previous observation of altered neural synaptic neurotransmission in the cerebellum of postnatal day (P) 14 α3-HT pups [32], our findings suggest that a chronic decrease in ASC content during brain development (before and after birth) may affect the formation of functional basal ganglia and cerebellar neural circuits. This idea is in line with evidence that adult onset of chronic diseases, such as hypertension and diabetes, depends in part on environmental influences that act from early in life [84]. We propose that neurologic disorders based on genetic mutations are no exception to the concept of "developmental origins of health and diseases".

We observed a significant decrease in the ASC content in the basal ganglia and cerebellum of peripartum α3-HT mice. The onset of symptoms of *ATP1A3*-related neurologic disorders is almost always related to physical and psychological stress. The signs and symptoms of RDP, or DYT12, commonly occur in adolescence or early adulthood and are triggered by stressors, such as excessive exercise (e.g., running track), febrile illness, pregnancy/delivery, and alcohol binges. The onset and manifestations of AHC are caused by psychological excitement, environmental stressors, excessive exercise, illness, and irregular sleep. Febrile illness and pregnancy are reported triggers of CAPOS (reviewed in [15]). Previously, we reported that following restraint stress, α3-HT mice showed walking disturbances, which is a typical symptom of cerebellar dysfunction [43]. Our analysis of α3-HT mice, showing that these mice usually have lower levels of ASC in the basal ganglia and cerebellum beginning in the peripartum period, may be clinically relevant. It is worth noting that a 2-fold increase in ASC content in the livers of pregnant SVCT2-HT mice compared to the livers of non-pregnant SVCT2-HT and SVCT2-WT (SVCT2[+/+]) mice has been observed [37]. The most likely explanation for this observation in dams is that ASC levels increase to meet the needs of the developing foetal brain [85]. In other words, the dam and foetus in the womb should indeed be push to the limit

in terms of the adequate function of ASC in the brain. The potential for ASC deficiency during human brain development [86] is magnified because humans lacks the capacity for *de novo* synthesis of ASC and thus are unable to upregulate ASC levels like mice. Although most typical diets are considered nutritionally replete with ASC, subclinical ASC deficiency has been identified in a number of populations, including pregnant and lactating women [87]. Even without triggering extreme deficiency including scurvy, it is possible that chronic maternal deficiency during pregnancy could be a risk factor for ASC deficiency in the foetus, which may be greatest in the basal ganglia and cerebellum. We consider it important to determine how the reduction in ASC content caused by physiological stress (pregnancy) may impact redox as well as other pathways, and eventually affect development and functional maturation in the foetal brain. We are now investing whether a sufficient supply of ASC from the diet beginning before birth through the adult stage affects various phenotypes, such as the threshold of migraine aura in α2-HT mice and the threshold of onset of dystonia in adult α3-HT mice.

## Acknowledgments

The authors thank Y. Suto, M. Akima, and K. Takase for technical assistance.

## Author Contributions

**Conceptualization:** Keiko Ikeda.

**Data curation:** Keiko Ikeda, Adriana A. Tienda, Fiona E. Harrison, Kiyoshi Kawakami.

**Formal analysis:** Keiko Ikeda, Adriana A. Tienda.

**Funding acquisition:** Keiko Ikeda.

**Investigation:** Keiko Ikeda.

**Methodology:** Keiko Ikeda.

**Project administration:** Keiko Ikeda.

**Resources:** Keiko Ikeda, Fiona E. Harrison, Kiyoshi Kawakami.

**Supervision:** Fiona E. Harrison.

**Validation:** Keiko Ikeda, Adriana A. Tienda, Fiona E. Harrison.

**Writing – original draft:** Keiko Ikeda.

**Writing – review & editing:** Keiko Ikeda, Fiona E. Harrison, Kiyoshi Kawakami.

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
