## [Decision Letter · Decision Letter 0]

7 Oct 2020

PONE-D-20-29560

Decreased Content of Ascorbic acid (Vitamin C) in the Brain of a Mouse Model of *ATP1A3*-related Neurologic Disorders

PLOS ONE

Dear Dr. Ikeda,

Thank you for submitting your manuscript to PLOS ONE. After careful consideration, we feel that it has merit but does not fully meet PLOS ONE’s publication criteria as it currently stands. Therefore, we invite you to submit a revised version of the manuscript that addresses the points raised during the review process.

Both reviewers showed great interests in the core findings of the manuscript that the mutant mice showed significant decreased ASC level. I also found the results fascinating and potentially opened a new avenue of research in the field. However, both reviewers also pointed out deficiencies of the manuscript that needs to be addressed before the manuscript can be published. Reviewer 1 requested further study of the heterozygous SVCT2 mice, in vitro study of SVCT2 cell culture, ouabain in vivo or cell culture experiment. Reviewer 1 also expressed concerns of off target effect of the CRISPR experiments. Reviewer 2 requested additional information on the expression levels of SVCT2. I do not see any conflicts between the comments of these two excellent reviewers. I believe these additional experiments will improve the manuscript if the animal and cell resources are readily available and can be completed in a timely manner.  

We look forward to receiving your revised manuscript.

Kind regards,

Yuqing Li, Ph.D.

Academic Editor

PLOS ONE

Journal Requirements:

3. Thank you for including your ethics statement:  "All experimental protocols described in the present study were carried out in a humane manner and in accordance with the institutional regulations of the “Animal Experiments and Fundamental Guideline for Proper Conduct of Animal Experiments and Related Activities in Academic Research Institutions” under the jurisdiction of the Japanese Ministry of Education, Culture, Sports, Science and Technology (MEXT) and the Animal Research Committees of Jichi Medical University, which operate in accordance with Law No. 105 of the Japanese Government for the care and use of laboratory animals. All efforts were made to minimize the number of animals used and their suffering.

Deep anaesthesia using inhaled isoflurane.".   

Please amend your current ethics statement to confirm that your named ethics committee specifically approved this study.

For additional information about PLOS ONE submissions requirements for ethics oversight of animal work, please refer to http://journals.plos.org/plosone/s/submission-guidelines#loc-animal-research  

5. Please amend either the title on the online submission form (via Edit Submission) or the title in the manuscript so that they are identical.

Reviewers' comments:

Reviewer's Responses to Questions

**Comments to the Author**

1. Is the manuscript technically sound, and do the data support the conclusions?

Reviewer #1: Partly

Reviewer #2: Yes

2. Has the statistical analysis been performed appropriately and rigorously? 

Reviewer #1: I Don't Know

Reviewer #2: Yes

3. Have the authors made all data underlying the findings in their manuscript fully available?

Reviewer #1: Yes

Reviewer #2: Yes

4. Is the manuscript presented in an intelligible fashion and written in standard English?

Reviewer #1: Yes

Reviewer #2: Yes

5. Review Comments to the Author

Reviewer #1: In this article Ikeda et al. compared the phenotypes of mouse fetuses with double homozygous knockout of Atp1a2 and Atp1a3 (α2α3-dKO) to those of fetuses with single knockout. They found:

1. The brain hemorrhage phenotype of α2α3-dKO was quite similar to that of fetuses with homozygous knockout of the gene encoding ascorbic acid (ASC) transporter SVCT2.

2. The α2α3-dKO brain showed significantly decreased levels of ASC compared with the single knockout mouse brain

3. ASC content in the basal ganglia and cerebellum was significantly lower in the adult Atp1a3 heterozygous knockout mouse (α3-HT) than in the wild-type mouse.

4. Significant decrease in ASC in the cerebellum of α3-HT mice in the peripartum period, during which mice are under physiological stress.

The authors then conclude that their observations indicate that the α2 and α3 subunits independently and additionally contribute to the ASC level in the fetal brain and that the α3 subunit contributes to the adult basal ganglia and cerebellum. They then propose that decreased ASC levels may affect neural network development and are linked to the pathophysiology of symptoms in ATP1A2 and related neurologic disorders.

Overall Comments:

This article generates intriguing data of very original hypotheses very worthy of investigation. However, it fails, with the experiments performed so far, to provide a coherent narrative supported by strong data. Substantially more work is needed to be able to support such a coherent narrative.

The demonstration of reduced levels of ASC does not necessarily indicate that this reduction is contributing significantly to the pathology or pathophysiology of ATP1A2 and ATP1A3 disease. To support this hypothesis of theirs they would need to demonstrate that such reductions in the absence of ATP1A2 and ATP1A3 mutations reproduce the phenotype and physiology of the heterozygous knock out mice. One way to do that is to study the heterozygous or some similar transgenic model of the ASC transporter determine the levels in such a model, demonstrate these are in the range of what they observed in their heterozygous knockouts and show phenotype and neurophysiological similarities. The comparisons of the homozygous severe phenotypes and double knockout, where as interesting, are generating severe phenotypes that may not be relevant to the actual disease model (heterozygous knock outs) or to the human condition. Of note, for example, is that patients with ATP1A3 disease have not been noted to have intracranial hemorrhage. Performing the above is important particularly that the authors did not find any decreases in malondialdehyde (MDA), and sulfhydryl (SH) which does not support their hypotheses and because they found that ASC levels were not decreased in cortex. It is known that mice and humans with ATP1A2 and ATP1A3 related disease do have cortical abnormalities. This argues against their hypotheses.

Additional experiments that also could support their hypotheses are in vitro studies to demonstrate in cell cultures the effect of the mutation on ASC transport.

Abstract and Introduction:

Those are too long and can be summarized.

Methods and study design:

The authors need to justify the relevance of why they used study double homozygous knockout mice if there is no human with this condition.

Which parts of the study were done in which university and what was the role of Adriana A. Tienda, and of Fiona E. Harrison

What were the Coefficient of variations and of the assays they performed.

What program was used for statistics?

The authors need to study the hippocampus, or justify why it was not studied, since in many studies it was shown that this structure has significant role in ATP1A3 mutant mice phenotype.

Results:

The authors did not show directly that Na/K pump is essential for ASC transporting. They have to use Na/K pump inhibitor (ouabain) or in vitro experiments in cell cultures to demonstrate this relationship.

Figure 3 indicated that the alpha 3 knockout adults have lower ASC levels than WT while figure 6 appears to indicate there is no difference. This needs to be clarified.

In Figure 1 there is no described sequence event though in they indicate in the text that the sequence is in that figure.

What is the role of ATP1A3 and ATP1A2 with abdominal wall rapture? Maybe there is off target mutation? How can the authors be sure that there is no off target mutation given that they used CRISPR technique? Sanger sequencing of targeted/investigated genes could be essential.

Regarding the statement “To examine whether physiological stress decreased brain vitamin C in ATP1A3 mouse model affects ASC levels in the brain, we measured ASC content in brain regions (cortex, basal ganglia, and cerebellum) of peripartum female mice that were either in full-term pregnancy or lactating within 12 hours of delivering a litter (Fig. 6).” Question regarding this experiments: How physiological stress performed? Or are they just referring to the stress of the peripartum period?

Discussion

The authors need to discuss the implications that in humans ATP1A2 non neuronal and emphasize even more that their double knock out mice do not model the human condition so they only raise the possibility that vitamin C may contribute to the pathophysiology of symptoms manifested in humans.

The discussion omits important implications and potential correlations with the human conditions and with the other related models that actually would indirectly support a possible relationship of ASC in ATP1A2 and ATP1A3 related disease. These need to be added. They include 1) ASC can block glutamatergic dysfunction (PMID: 7816935) and ATP1A2 and ATP1A3 disease models (PMID: 30071271 PMID: 19666602 PMID: 25523819 PMID: 27445835). 2) ASC enhances GABA function (PMID 21715633) and GABA dysfunction is prominent in ATP1A3 related disease models (PMID : 29889309). 3) ASC deficiency can produce congenital brain anomalies with neonatal manifestation (PMID: 25487414) and ATP1A2 and ATP1A3 related disease can manifest similarly in humans (PMID: 31608932, PMID: 32802951, PMID: 31484714). 4) There most remarkable findings were in the cerebellum, this is consistent with prior data about the importance of ASC in formation of the cerebellum (PMID: 4830116). It also consistent with data from human studies that shows that ATP1A3 related disease frequently shows congenital and subsequent cerebellar hypoplasia and atrophy (PMID: 24651393, PMID: 32115366, PMID: 31950366).

The statement “Since the post-birth deaths of these Decreased brain vitamin C in ATP1A3 mouse model (Page 25) mice were likely due to the lack of spontaneous respiratory neural activity, in addition to severe hemorrhage inside the brainstem, it is suggested that these mice exhibit defects in the respiratory neural network [37,47], similar to Na + , K+ -ATPase α2 and α3 homozygous knockout mice (α2N-KO and α3-KO)” is purely conjectural. They should provide data to support that. They did not measure neural activity so how do they know that their observation are due to lack of spontaneous respiratory neural activity, how do they know that this is not due to primary cardiac or epileptic activity causing secondary apnea?

Minor Points

Abbreviations need to be included at time of first mention of compound (e.g of malondialdehyde (MDA).

References are needed for the following statements: “Because many neurological diseases are characterized by decreased brain vitamin C (need reference(s)… in ATP1A3 mouse.”; And “Alternatively, the function of the α2 subunit may be compensated by the ubiquitously expressed α1 subunit during brain development”.

The quality of Figure 5 needs to be improved since some words are difficult to read.

Reviewer #2: The main finding reported in the manuscript is reduced levels of ascorbic acid (vitamin C) in NKA alpha2 KO, NKA alpha3 KO and NKA alpha2/alpha3 double KO whole foetal mouse brain, in addition to basal ganglia and cerebellum from 2- to 4-month-old adult NKA alpha3 heterozygous mice. This is a novel and interesting finding. The manuscript is generally well written and presented. I do have several minor points of criticism though, which prevent the manuscript from being suitable for publication in its current form.

1. The Introduction refers to heterozygous mutations in ATP1A2 and ATP1A3 that are related to human neurological disorders, but the main types of mutation involved are not specified. Given the subjects of the study, readers might assume that deletions or nonsense mutations of ATP1A2 and ATP1A3 were the primary cause of the disorders mentioned, but this is not correct. The authors should clarify that it is actually missense mutations that are the primary cause.

2. On page 95, the authors should clarify what is meant by "homozygous missense/truncating (deletion)". Missense mutations are different from truncating mutations, which could be nonsense mutations.

3. In the Materials and Methods and in Table 1, the authors should clarify the identities of the mouse lines used. Specifically, it would be helpful if they used the official Mouse Genome Informatics nomenclature, at least once, as follows:

http://www.informatics.jax.org/allele/MGI:2664907

Atp1a2tm1Kwk

Atp1a2-, C-KO

http://www.informatics.jax.org/allele/MGI:3522421

Atp1a2tm2Kwk

N-KO

http://www.informatics.jax.org/allele/MGI:5572809

Atp1a3tm1.1Kwk

4. In the Materials and Methods, the diet fed to the mice is described merely as "Food (a standard pelleted rodent diet containing 4 mg ASC in 100 g pellet)". This is an inadequate description. The composition of the mouse diet should be stated more thoroughly.

5. In the sub-section headed Creation of Slc23a2-KO mice, the authors should rephrase and clarify the following sentence, as it is not clear to this reviewer: "Because all the latter were found to be random integrations in non-homologous chromosomes, we used only Slc23a2-KO for the subsequent analyses."

6. In the Results, the authors focus on sodium-dependent vitamin C transporter 2 (SVCT2), encoded by Slc23a2. It is not shown whether NKA alpha2 KO, NKA alpha3 KO and NKA alpha2/alpha3 double KO whole foetal mouse brain has wild-type like levels of SVCT2. It would be helpful if the authors used IHC, western blotting or RT-PCR to examine the levels of SVCT2 or Slc23a2 in whole foetal brain from these mouse lines. If levels of SVCT2 or Slc23a2 are unaltered compared with wild-type mice, this would support the authors' conclusion that dysfunction of NKA alpha2 or alpha3 is responsible for the low ascorbic acid levels.

7. In the Discussion, it is stated that seizures are exaggerated by low ASC in a mouse model. The authors should elaborate by stating which mouse model and human disease modelled they are referring to.

8. In the Discussion, it would be helpful if the authors would explain how they might, in future, test their hypothesis that "decreased levels of ASC relate to ATP1A2- and ATP1A3-related neurologic disorders".

9. The title of the manuscript (Decreased Content of Ascorbic acid (Vitamin C) in the Brain of Mouse Models of Na+, K+-ATPase-related Neurologic Disorders) does not state that the subjects of the study are actually NKA alpha2 and NKA alpha3 KO mice. A more informative title would be: 'Decreased Content of Ascorbic acid (Vitamin C) in the Brain of Knockout Mouse Models of Na+, K+-ATPase-related Neurologic Disorders'. This is important because the vast majority of NKA-related disorders are caused by missense mutations rather than deletions or nonsense mutations of ATP1A2 and ATP1A3.

10. The Introduction refers to 'a small number of AHCs'. This should be rephrased to clarify what this means.

11. There are a few typographical issues:

- 'Na+, K+-ATPase' is more commonly written without a space between the , and K: Na+,K+-ATPase.

- 'the homozygous knockout neonatal brain (Atp1a3-/-) (Table 1) [32,43],.' - delete the , after ].

- 'The ASC content in the cortex and basal ganglia also decreased in the peripartum α3-HT mice' - change to 'The ASC content in the cortex and basal ganglia was also decreased in the peripartum of α3-HT mice'.

6. PLOS authors have the option to publish the peer review history of their article (what does this mean?). If published, this will include your full peer review and any attached files.

Reviewer #1: No

Reviewer #2: No

---

## [Author Response · Author response to Decision Letter 0]

17 Dec 2020

December 17th, 2020

We would like to extend our sincere thanks to the reviewers for their careful evaluation of our manuscript. We have responded to each of the individual queries and concerns raised below.

In addition to the individual changes requested, we have also rerun the statistical analyses using exactly the same raw data but using different statistical program from that has been used previously (new analyses performed by Dr. Harrison using Graphpad Prism). We are now able to report that the significant decrease of ASC during peripartum mice is observed not only the cerebellum, but also in the basal ganglia. We are glad to have had to opportunity to correct this oversight and we changed the relevant parts of our results and conclusions accordingly.

Sincerely, Dr. K. Ikeda, 

on behalf of all of the authors

5. Review Comments to the Author

Reviewer #1: In this article Ikeda et al. compared the phenotypes of mouse fetuses with double homozygous knockout of Atp1a2 and Atp1a3 (α2α3-dKO) to those of fetuses with single knockout. They found:

1. The brain hemorrhage phenotype of α2α3-dKO was quite similar to that of fetuses with homozygous knockout of the gene encoding ascorbic acid (ASC) transporter SVCT2.

2. The α2α3-dKO brain showed significantly decreased levels of ASC compared with the single knockout mouse brain

3. ASC content in the basal ganglia and cerebellum was significantly lower in the adult Atp1a3 heterozygous knockout mouse (α3-HT) than in the wild-type mouse.

4. Significant decrease in ASC in the cerebellum of α3-HT mice in the peripartum period, during which mice are under physiological stress.

The authors then conclude that their observations indicate that the α2 and α3 subunits independently and additionally contribute to the ASC level in the fetal brain and that the α3 subunit contributes to the adult basal ganglia and cerebellum. They then propose that decreased ASC levels may affect neural network development and are linked to the pathophysiology of symptoms in ATP1A2 and related neurologic disorders.

Overall Comments:

This article generates intriguing data of very original hypotheses very worthy of investigation. However, it fails, with the experiments performed so far, to provide a coherent narrative supported by strong data. Substantially more work is needed to be able to support such a coherent narrative.

The demonstration of reduced levels of ASC does not necessarily indicate that this reduction is contributing significantly to the pathology or pathophysiology of ATP1A2 and ATP1A3 disease.

R1.1. To support this hypothesis of theirs they would need to demonstrate that such reductions in the absence of ATP1A2 and ATP1A3 mutations reproduce the phenotype and physiology of the heterozygous knock out mice. One way to do that is to study the heterozygous or some similar transgenic model of the ASC transporter determine the levels in such a model, demonstrate these are in the range of what they observed in their heterozygous knockouts and show phenotype and neurophysiological similarities. 

Thank you very much for this valuable suggestion. In the revised manuscript, we have added examples of mice which show neurological deficits with decreased level of ASC in the section “Insights into the decrease of ASC in the brain and ATP1A2- and ATP1A3-related neurologic disorders” of Discussion (lines 414-422).

As for Atp1a3 heterozygous knockout mice used in this paper, we added the sentence below in the Result section (lines 352-353). Unfortunately, we have not measured ASC content in these two mice.

Actually, we observed transient dystonic spells in female α3-HT mice after delivery (n=2, data not shown).

R1.2. The comparisons of the homozygous severe phenotypes and double knockout, where as interesting, are generating severe phenotypes that may not be relevant to the actual disease model (heterozygous knock outs) or to the human condition. Of note, for example, is that patients with ATP1A3 disease have not been noted to have intracranial hemorrhage.

As the reviewer pointed out, there is no reports of double knockout human condition. Our original purpose in analysing double knockout mice was to uncover whether there are distinct or redundant roles in Atp1a2 and Atp1a3 during development, since both are found to be expressed in neurons in late foetal stage. If they had redundant roles, then, double homozygous mice should show similar phenotypes to that of single knockout of Atp1a2 or Atp1a3 (immediate death after birth and no obvious anatomical defects of appearance). Instead, we observed apparent brain haemorrhage in double knockout animals, that has not been observed in either of the single knockouts. This novel phenotype led to the discovery of the critical role of Atp1a2 and Atp1a3 in the transport of ASC into the brains during development and in adulthood. Therefore, although the double knockout is not a direct reflection of human disease, this approach including analyses of double knockout mice permitted novel insight into the roles of these transporters including identifying a previously unknown function.

R1.3. Performing the above is important particularly that the authors did not find any decreases in malondialdehyde (MDA), and sulfhydryl (SH) which does not support their hypotheses and because they found that ASC levels were not decreased in cortex. It is known that mice and humans with ATP1A2 and ATP1A3 related disease do have cortical abnormalities. This argues against their hypotheses.

It has been postulated that conventional brain MRI and CT are not useful for diagnosing ATP1A3-related neurological disorders, since they usually show no specific abnormal findings in patients with RDP and AHC at least early in the course of the disease (Brashear et al., https://www.ncbi.nlm.nih.gov/books/NBK1115/). However, as the reviewer points out, cortical abnormalities in ATP1A2/ATP1A3-neurologic disorder are recently getting attention. We followed the reviewer’s advice and have added the following text to address these ideas (lines 439-441).

Very recently, it was found that patients with AHC frequently show congenital and subsequent cerebellar hypoplasia and atrophy [80-82] and that FHM2 patients show biochemical changes in the cerebellum, as measured by proton-nuclear magnetic resonance spectroscopy [83].

To address our failure to observe significant increases in global oxidative stress in α3-HT mice we have added the following text to the Results section (lines 339-342).

The results highlight that the extent of ASC decrease in α3-HT mice did not drive increased oxidative stress and further suggest that the ASC decrease was not generated by an excess of reactive oxygen species (ROS). Instead we propose that the change was due to some other facet of ASC transporter function coupled with the α3 subunit of Na+,K+-ATPase. 

R1.4. Additional experiments that also could support their hypotheses are in vitro studies to demonstrate in cell cultures the effect of the mutation on ASC transport.

For response, please see R1.11 Results.

R1.5. Abstract and Introduction: Those are too long and can be summarized.

We have shortened the Abstract (word counts decreased from 395 to 299) and the Introduction (word counts reduced from 775 to 755).

R1.6. Methods and study design: The authors need to justify the relevance of why they used study double homozygous knockout mice if there is no human with this condition.

It is reasonably common practice to use knockout mice, or as in this case double knockouts, when the human conditions is only a single mutation. Mice are not perfect replicas of the human condition and multiple compensatory mechanisms may be in place. However, this approach still permits us to learn about the basic biology of these transporters and the mice models are not intended to fully recapitulate a specific human clinical phenotype. 

Following the reviewer’s comment, to clarify why we study double homozygous knockout mice, we describe as follows in the Introduction (lines 82-87).

We hypothesized that developmental dysfunction of the α2 and/or α3 subunits might be a precondition for symptomatic manifestation of these neurologic disorders. Therefore, it is important to determine whether the α2 and α3 subunits, both of which are expressed in developmental neurons, play complementary or independent roles during brain development. To address this issue, we crossed Atp1a2/Atp1a3 double heterozygous knockout mice to generate Atp1a2/Atp1a3 double homozygous knockout foetuses.

R1.7. Which parts of the study were done in which university and what was the role of Adriana A. Tienda, and of Fiona E. Harrison

Dr. Harrison and Ms Tienda performed the MDA and sulfhydryl assays along with confirmatory ASC measurements in the Atp1a3 and SVCT2/Atp1a3 HT models. Dr. Harrison collaborated with Dr. Ikeda to design some of the experiments and actively contributed to review and edit of the manuscript. All Statistical analyses and related Figures were conducted by Dr. Harrison in the revised manuscript. 

R1.8. What were the Coefficient of variations and of the assays they performed.

Coefficients of variance are not typically included with these types of analyses and we have chosen not to include this information in the revision to avoid confusion. However, we confirmed that the relative variability was similar among groups analysed and thus data were appropriate for parametric analyses. Data are now displayed to show individual values as well as group means and spread which should make this clearer to the reader and also mean that numerical detail of the coefficient of variability is not necessary. 

R1.9. What program was used for statistics?

For the Re-analysis we used Prism v6 exclusively as described in the material and method section.

R1.10. The authors need to study the hippocampus, or justify why it was not studied, since in many studies it was shown that this structure has significant role in ATP1A3 mutant mice phenotype.

Thank you very much for valuable comments. The cortex prepared in this manuscript include hippocampus. As reviewer point out, it should be very interesting and important to check the ASC content only in hippocampus. We are planning to do this in the future project. To clarify this point for readers, we add “ including the hippocampus” in the section of “Measurement of ASC content” and “Measurement of MDA and SH levels” in the Materials and Methods (lines 131 and 166).

The whole brains or dissected regions of the brain (the cortex including the hippocampus, basal ganglia, and cerebellum) were immediately removed from …

Frozen brain regions (the cortex including hippocampus, basal ganglia, and cerebellum) prepared….

R1.11. Results: The authors did not show directly that Na/K pump is essential for ASC transporting. They have to use Na/K pump inhibitor (ouabain) or in vitro experiments in cell cultures to demonstrate this relationship.

SVCT2 is one of secondary active transporters which transports ASC across the cell membrane utilizing energy in other forms than ATP (Annual Review of Nutrition Vol. 25:105-125, 2005). This energy comes from the electrochemical gradient of Na+. The formation of the electrochemical gradient of Na+ is made by the primary active transporter, Na+,K+-ATPase (Na pump). There are no other pumps to create Na+ gradient in mammalian cell membranes (Guyton & Hall, Medial Physiology). Previously, we have shown Na+,K+-ATPase dependency of glutamate and GABA transports using primary culture cells prepared from knockout foetuses (+/+, +/-, -/-) with/without ouabain (Ikeda et al, 2003, Fig.6). There are several challenges to performing the suggested experiments in primary cell culture system. Firstly, cultured cells are usually heterogeneous and contain some mix of neurons, glial cells, and fibroblasts which could impact results. Second, ouabain inhibits all of alpha subunits (alpha1,2,3) of Na+,K+-ATPase and alpha1 is expressed in almost all cells. Lastly, SVCT2 is known to be transiently expressed in primary astrocytes in culture which is not the case in vivo, which strongly suggests that regulation of SVCT2 in vitro does not perfectly mirror the in vivo situation. Therefore, we think experiments using cultured cells are not essential to our conclusion, although we might be able to confirm functional coupling between Na pump and SVCT2 with/without ouabain with primary cultures at some point in the future.

R1.12. Figure 3 indicated that the alpha 3 knockout adults have lower ASC levels than WT while figure 6 appears to indicate there is no difference. This needs to be clarified.

All data have been reanalysed to confirm effects for this revised manuscripts and significant differences are clearly labelled in each Figure.

R1.13. In Figure 1 there is no described sequence event though in they indicate in the text that the sequence is in that figure.

The Figure referred to is now Figure 3. This Figure itself shows the overall scheme used to generate the mice, and is not necessary to referred in the Material and methods section that comes before the Result section. Following reviewer’s pointing out, we deleted the refer (Fig 3) from the Material and methods section. 

R1.14. What is the role of ATP1A3 and ATP1A2 with abdominal wall rapture? Maybe there is off target mutation? How can the authors be sure that there is no off target mutation given that they used CRISPR technique? Sanger sequencing of targeted/investigated genes could be essential.

Thank you very much for valuable comments. Because knockout mice of Atp1a2 and Atp1a3 used in this manuscript are made by conventional homologous recombination technique, not CRISPR technique (Ikeda et al., 2003; 2004; 2014), there are not expected to be off target effects as would be obtained through use of CRISPR technology. 

As for abdominal wall rupture, this question is currently under investigation by a colleague using recently established techniques in Dr. Kawakami’s lab to understand the molecular mechanism of abdominal wall rupture. (reference: https://www.ncbi.nlm.nih.gov/pmc/articles/PMC6215434/). In general, abdominal rapture phenotype is frequently observed in many gene-manipulating mice and may be caused by multiple factors. 

R1.15. Regarding the statement “To examine whether physiological stress decreased brain vitamin C in ATP1A3 mouse model affects ASC levels in the brain, we measured ASC content in brain regions (cortex, basal ganglia, and cerebellum) of peripartum female mice that were either in full-term pregnancy or lactating within 12 hours of delivering a litter (Fig. 6).” Question regarding this experiments: How physiological stress performed? Or are they just referring to the stress of the peripartum period?

We did not add additional stress to these female mice. We considered that peripartum period as physiological stress for two reasons. 1. Childbirth is reported as a trigger of manifesting neurologic symptoms in ATP1A3-related disorders in humans (Brashear et al., https://www.ncbi.nlm.nih.gov/books/NBK1115/). 2. We observed dystonic spell in female mice after delivery. We describe this point in the revised manuscript as follows (lines 352-353).

Actually, we observed transient dystonic spells in female α3-HT mice after delivery (n=2, data not shown).

R1.16. The authors need to discuss the implications that in humans ATP1A2 non neuronal and emphasize even more that their double knock out mice do not model the human condition so they only raise the possibility that vitamin C may contribute to the pathophysiology of symptoms manifested in humans.

As for the neuronal expression of α2, we describe the following sentence in the Introduction (lines 55-57).

Therefore, it is widely accepted that α3 is the neuronal type and α2 is the glial type. Contrary to this idea, we and others have reported that in mice, α2 is also expressed in neurons at birth [6,7].

According the comment, we add the below sentence in the Result section to clarify this point (lines 252-253).

At present, there have been no reports of patients with double mutations in ATP1A2 and ATP1A3.

R1.17. The discussion omits important implications and potential correlations with the human conditions and with the other related models that actually would indirectly support a possible relationship of ASC in ATP1A2 and ATP1A3 related disease. These need to be added. They include 1) ASC can block glutamatergic dysfunction (PMID: 7816935) and ATP1A2 and ATP1A3 disease models (PMID: 30071271 PMID: 19666602 PMID: 25523819 PMID: 27445835). 2) ASC enhances GABA function (PMID 21715633) and GABA dysfunction is prominent in ATP1A3 related disease models (PMID : 29889309). 3) ASC deficiency can produce congenital brain anomalies with neonatal manifestation (PMID: 25487414) and ATP1A2 and ATP1A3 related disease can manifest similarly in humans (PMID: 31608932, PMID: 32802951, PMID: 31484714). 4) There most remarkable findings were in the cerebellum, this is consistent with prior data about the importance of ASC in formation of the cerebellum (PMID: 4830116). It also consistent with data from human studies that shows that ATP1A3 related disease frequently shows congenital and subsequent cerebellar hypoplasia and atrophy (PMID: 24651393, PMID: 32115366, PMID: 31950366).

Thank you very much for thoughtful advice. According the comments, we add all of the references suggested above in the revised manuscript (Two of them have been appeared in the pre-revised manuscript).

PMID: 7816935, [60]; PMID: 30071271, [57]; PMID: 19666602, [54]; PMID: 25523819, [55]; PMID: 27445835, [30]; PMID: 21715633, [61]; PMID: 29889309, [56]; PMID: 25487414, [67]; PMID: 31608932, [28]; PMID: 32802951, [82]; PMID: 31484714, [77]; PMID: 4830116, [86]; PMID: 24651393, [83]; PMID: 32115366, [80]; PMID: 31950366, [81].

R1.18. The statement “Since the post-birth deaths of these Decreased brain vitamin C in ATP1A3 mouse model (Page 25) mice were likely due to the lack of spontaneous respiratory neural activity, in addition to severe hemorrhage inside the brainstem, it is suggested that these mice exhibit defects in the respiratory neural network [37,47], similar to Na + , K+ -ATPase α2 and α3 homozygous knockout mice (α2N-KO and α3-KO)” is purely conjectural. They should provide data to support that. They did not measure neural activity so how do they know that their observation are due to lack of spontaneous respiratory neural activity, how do they know that this is not due to primary cardiac or epileptic activity causing secondary apnea?

There is the description about primary cause of death as “mice died within a few minutes of birth with respiratory failure” in the first published paper about SVT2-KO (Sotiriou et al, 2002 Nat Med), although there is no experimental data shown in that manuscript. We also consider this point is very important as the reviewer points out, we are now investigating SVCT2-KO with our refined system to investigate respiratory neural network function using en bloc preparation. We will publish the results in the near future. We mentioned above in the Discussion (lines 385-388).

R1.19. Minor Points: Abbreviations need to be included at time of first mention of compound (e.g of malondialdehyde (MDA).

Because we describe its abbreviation in the material and method section which is before the result section, we did not rephrased. However, as reviewer’s pointed out, we also described in the result section in the revised manuscript.

R1.20. References are needed for the following statements: “Because many neurological diseases are characterized by decreased brain vitamin C (need reference(s)… in ATP1A3 mouse.”; And “Alternatively, the function of the α2 subunit may be compensated by the ubiquitously expressed α1 subunit during brain development”.

We rewrote and added appropriate references as suggested above in the Discussion. We added references about expression of α1 subunit [64,65].

R1.21. The quality of Figure 5 needs to be improved since some words are difficult to read.

All of the figures have been remade for this submission. 

Reviewer #2: The main finding reported in the manuscript is reduced levels of ascorbic acid (vitamin C) in NKA alpha2 KO, NKA alpha3 KO and NKA alpha2/alpha3 double KO whole foetal mouse brain, in addition to basal ganglia and cerebellum from 2- to 4-month-old adult NKA alpha3 heterozygous mice. This is a novel and interesting finding. The manuscript is generally well written and presented. I do have several minor points of criticism though, which prevent the manuscript from being suitable for publication in its current form.

R2.1. The Introduction refers to heterozygous mutations in ATP1A2 and ATP1A3 that are related to human neurological disorders, but the main types of mutation involved are not specified. Given the subjects of the study, readers might assume that deletions or nonsense mutations of ATP1A2 and ATP1A3 were the primary cause of the disorders mentioned, but this is not correct. The authors should clarify that it is actually missense mutations that are the primary cause.

Thank you very much for valuable comment. We added this point to the Results section as follows (lines 242-245).

Although the primary causes of the human ATP1A2- and ATP1A3-related neurologic disorders described above are missense mutations, these heterozygous knockout (deletion) mice have been shown to exhibit the phenotypes of FHM2 (Atp1a2+/-) [42] and RDP but not AHC (Atp1a3+/-) [43,44].

R2.2. On page 95, the authors should clarify what is meant by "homozygous missense/truncating (deletion)". Missense mutations are different from truncating mutations, which could be nonsense mutations.

Thank you very much for valuable comment. We corrected this (lines 76-79).

The most severe clinical presentation of the ATP1A2 variant/mutation is homozygous truncation of ATP1A2, which results in severe brain developmental anomalies followed by neonatal or early infantile death [28,29].

R2.3. In the Materials and Methods and in Table 1, the authors should clarify the identities of the mouse lines used. Specifically, it would be helpful if they used the official Mouse Genome Informatics nomenclature, at least once, as follows:

http://www.informatics.jax.org/allele/MGI:2664907

Atp1a2tm1Kwk

Atp1a2-, C-KO

http://www.informatics.jax.org/allele/MGI:3522421

Atp1a2tm2Kwk

N-KO

http://www.informatics.jax.org/allele/MGI:5572809

Atp1a3tm1.1Kwk

Thank you very much for valuable comment. We added the nomenclatures above in the revised Table 1.

R2.4. In the Materials and Methods, the diet fed to the mice is described merely as "Food (a standard pelleted rodent diet containing 4 mg ASC in 100 g pellet)". This is an inadequate description. The composition of the mouse diet should be stated more thoroughly.

We describe composition of diet in the Materials and Methods. We also found and corrected a mistake regarding the content of ASC which was but 25 mg in 100 g pellet (not 4g as previously stated). We are very pleased to have had to opportunity to correct this prior to publication. 

R2.5. In the sub-section headed Creation of Slc23a2-KO mice, the authors should rephrase and clarify the following sentence, as it is not clear to this reviewer: "Because all the latter were found to be random integrations in non-homologous chromosomes, we used only Slc23a2-KO for the subsequent analyses."

Thank you very much for valuable comment. We added the following sentence (lines 219-221).

Because all the latter were found to be random integrations in non-homologous chromosomes, we used only Slc23a2-KO mice for the subsequent analyses.

R2.6. In the Results, the authors focus on sodium-dependent vitamin C transporter 2 (SVCT2), encoded by Slc23a2. It is not shown whether NKA alpha2 KO, NKA alpha3 KO and NKA alpha2/alpha3 double KO whole foetal mouse brain has wild-type like levels of SVCT2. It would be helpful if the authors used IHC, western blotting or RT-PCR to examine the levels of SVCT2 or Slc23a2 in whole foetal brain from these mouse lines. If levels of SVCT2 or Slc23a2 are unaltered compared with wild-type mice, this would support the authors' conclusion that dysfunction of NKA alpha2 or alpha3 is responsible for the low ascorbic acid levels.

Thank you very much for the valuable comment. We agree with this point; therefore, we performed reverse transcription-qPCR experiments and have now included the result in Figure 2B, stating that the expression level of Slc23a2 is not altered among the wild-type and all of other knockout mouse lines as below (lines 299-300).

We found that there was no significant difference in the mRNA expression of Slc23a2 among genotypes (Fig 2B).

R2.7. In the Discussion, it is stated that seizures are exaggerated by low ASC in a mouse model. The authors should elaborate by stating which mouse model and human disease modelled they are referring to.

Thank you very much for valuable comment. We add references for the mouse models (lines 423-425).

Seizures are commonly observed in ATP1A2- and ATP1A3-related neurological disorders [15,77], and in a mouse model of Alzheimer’s diseases, seizures are exaggerated by low ASC levels [74,75].

R2.8. In the Discussion, it would be helpful if the authors would explain how they might, in future, test their hypothesis that "decreased levels of ASC relate to ATP1A2- and ATP1A3-related neurologic disorders".

This is an important point. We have added following sentence in the last part of the Discussion (lines 474-476).

We are now investing whether a sufficient supply of ASC from the diet beginning before birth through the adult stage affects various phenotypes, such as the threshold of migraine aura in α2-HT mice and the threshold of onset of dystonia in adult α3-HT mice.

R2.9. The title of the manuscript (Decreased Content of Ascorbic acid (Vitamin C) in the Brain of Mouse Models of Na+, K+-ATPase-related Neurologic Disorders) does not state that the subjects of the study are actually NKA alpha2 and NKA alpha3 KO mice. A more informative title would be: 'Decreased Content of Ascorbic acid (Vitamin C) in the Brain of Knockout Mouse Models of Na+, K+-ATPase-related Neurologic Disorders'. This is important because the vast majority of NKA-related disorders are caused by missense mutations rather than deletions or nonsense mutations of ATP1A2 and ATP1A3.

This is a valuable point. We have revised the title following the reviewer’s advice.

R2.10. The Introduction refers to 'a small number of AHCs'. This should be rephrased to clarify what this means.

Thank you very much for pointing this out. We rephrased the statement as follows (lines 74-75).

and AHC, although the number of cases of AHC caused by ATP1A2 is small compared to that caused by ATP1A3 [24,25].

R2.11. There are a few typographical issues:

- 'Na+, K+-ATPase' is more commonly written without a space between the , and K: Na+,K+-ATPase.

- 'the homozygous knockout neonatal brain (Atp1a3-/-) (Table 1) [32,43],.' - delete the , after ].

- 'The ASC content in the cortex and basal ganglia also decreased in the peripartum α3-HT mice' - change to 'The ASC content in the cortex and basal ganglia was also decreased in the peripartum of α3-HT mice'.

Thank you very much for highlighting these issues. We have corrected each of these occurrences and rephrased parts of the revised manuscript as appropriate.

---

## [Decision Letter · Decision Letter 1]

20 Jan 2021

PONE-D-20-29560R1

Decreased content of ascorbic acid (vitamin C) in the brain of knockout mouse models of Na+,K+-ATPase-related neurologic disorders

PLOS ONE

Dear Dr. Ikeda,

Thank you for submitting your manuscript to PLOS ONE. After careful consideration, we feel that it has merit but does not fully meet PLOS ONE’s publication criteria as it currently stands. Therefore, we invite you to submit a revised version of the manuscript that addresses the points raised during the review process.

We look forward to receiving your revised manuscript.

Kind regards,

Yuqing Li, Ph.D.

Academic Editor

PLOS ONE

Reviewers' comments:

Reviewer's Responses to Questions

**Comments to the Author**

1. If the authors have adequately addressed your comments raised in a previous round of review and you feel that this manuscript is now acceptable for publication, you may indicate that here to bypass the “Comments to the Author” section, enter your conflict of interest statement in the “Confidential to Editor” section, and submit your "Accept" recommendation.

Reviewer #2: (No Response)

2. Is the manuscript technically sound, and do the data support the conclusions?

Reviewer #2: Yes

3. Has the statistical analysis been performed appropriately and rigorously? 

Reviewer #2: Yes

4. Have the authors made all data underlying the findings in their manuscript fully available?

Reviewer #2: Yes

5. Is the manuscript presented in an intelligible fashion and written in standard English?

Reviewer #2: Yes

6. Review Comments to the Author

Reviewer #2: All of my comments have been adequately addressed apart from my point on the ambiguity of the following passage: "Surviving 1-cell embryos were transferred into the oviducts of pseudopregnant ICR females. Slc23a2-KO (strain deposited in the RIKEN BioResource Research Center) and Slc23a2-floxed mice were generated as previously described [40,41]. Because all the latter were found to be random integrations in non-homologous chromosomes, we used only Slc23a2-KO mice for the subsequent analyses." Please clarify in the manuscript what is being referred to by "all the latter".

7. PLOS authors have the option to publish the peer review history of their article (what does this mean?). If published, this will include your full peer review and any attached files.

Reviewer #2: No

---

## [Author Response · Author response to Decision Letter 1]

21 Jan 2021

Response to Reviewers

January 21th, 2021

We would like to extend our sincere thanks to the reviewers for their careful evaluation of our manuscript. We have responded to the query raised below. We are glad to have had to opportunity to correct this sentence. Thank you very much.

Sincerely, Dr. K. Ikeda, 

on behalf of all of the authors

Reviewer #2: All of my comments have been adequately addressed apart from my point on the ambiguity of the following passage: "Surviving 1-cell embryos were transferred into the oviducts of pseudopregnant ICR females. Slc23a2-KO (strain deposited in the RIKEN BioResource Research Center) and Slc23a2-floxed mice were generated as previously described [40,41]. Because all the latter were found to be random integrations in non-homologous chromosomes, we used only Slc23a2-KO mice for the subsequent analyses." Please clarify in the manuscript what is being referred to by "all the latter".

Thank you very much for this valuable comment. In the re-revised manuscript, we have clarified "all the latter" to “all the Slc23a2-floxed mice”.

---

## [Decision Letter · Decision Letter 2]

25 Jan 2021

Decreased content of ascorbic acid (vitamin C) in the brain of knockout mouse models of Na+,K+-ATPase-related neurologic disorders

PONE-D-20-29560R2

Dear Dr. Ikeda,

We’re pleased to inform you that your manuscript has been judged scientifically suitable for publication and will be formally accepted for publication once it meets all outstanding technical requirements.

Kind regards,

Yuqing Li, Ph.D.

Academic Editor

PLOS ONE

Additional Editor Comments (optional):

Reviewers' comments:

Reviewer's Responses to Questions

**Comments to the Author**

1. If the authors have adequately addressed your comments raised in a previous round of review and you feel that this manuscript is now acceptable for publication, you may indicate that here to bypass the “Comments to the Author” section, enter your conflict of interest statement in the “Confidential to Editor” section, and submit your "Accept" recommendation.

Reviewer #2: All comments have been addressed

2. Is the manuscript technically sound, and do the data support the conclusions?

Reviewer #2: (No Response)

3. Has the statistical analysis been performed appropriately and rigorously? 

Reviewer #2: (No Response)

4. Have the authors made all data underlying the findings in their manuscript fully available?

Reviewer #2: (No Response)

5. Is the manuscript presented in an intelligible fashion and written in standard English?

Reviewer #2: (No Response)

6. Review Comments to the Author

Reviewer #2: (No Response)

7. PLOS authors have the option to publish the peer review history of their article (what does this mean?). If published, this will include your full peer review and any attached files.

Reviewer #2: No

---

## [Editor Report · Acceptance letter]

28 Jan 2021

PONE-D-20-29560R2 

Decreased content of ascorbic acid (vitamin C) in the brain of knockout mouse models of Na^+^,K^+^-ATPase-related neurologic disorders 

Dear Dr. Ikeda:

I'm pleased to inform you that your manuscript has been deemed suitable for publication in PLOS ONE. Congratulations! Your manuscript is now with our production department. 

Kind regards, 

on behalf of

Dr. Yuqing Li 

Academic Editor

PLOS ONE